# BioFormer: Rethinking Cross-Subject Generalization via Spectral Structural Alignment in Biomedical Time-Series

Guikang Du[1]  Haoran Li[* 2]  Xinyu Liu[1]  Zhibo Zhang[1]  Xiaoli Gong[2]  Jin Zhang[2]

## Abstract

Cross-subject generalization in biomedical time-series aims to learn representations that generalize to unseen subjects while suppressing subject-specific variability. Most existing methods implicitly suppress the variability through model building or subject adversarial learning, but rarely model it explicitly. We introduce **spectral drift** as a new perspective to characterize subject specific variability. Specifically, BTS signals under the same label often share consistent oscillatory structure, yet exhibit subject-dependent magnitude or phase shifts in specific frequency components, which we interpret as subject-specific variability. Building on this insight, we propose *BioFormer*. At its core is a Frequency-Band Alignment Module (FBAM) that generates band-wise modulation factors from the spectral distribution and adaptively adjusts amplitude and phase to align spectral structure, thereby mitigating variability. We further pair FBAM with Sample Conditional Layer Normalization, which infers normalization parameters from intrinsic signal statistics rather than subject identity, stabilizing cross-subject representations. Extensive experiments on six datasets demonstrate that BioFormer outperforms 12 baselines, yielding absolute F1-score improvements of 6%.

## 1. Introduction

Biomedical time-series (BTS) are temporally ordered measurements of physiological signals such as electroencephalo-

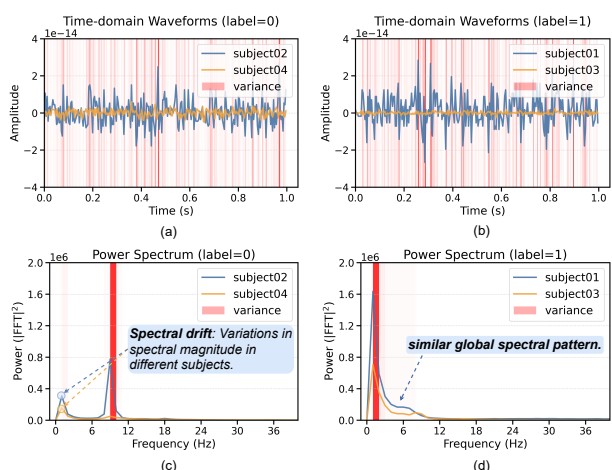

*Figure 1.* **Illustration of spectral drift in biomedical signal analysis.** While time-domain waveforms vary substantially across subjects, samples sharing the same label exhibit similar global spectral patterns; inter-subject variability is primarily manifested as band-wise magnitude and phase drift (data from the APAVA Dataset).

gram (EEG) and electrocardiogram (ECG) (Jovic et al., 2025; Cascarano et al., 2023; Liu et al., 2023). Across physiological and pathological conditions, these signals exhibit distinct and structured temporal–spectral patterns that arise from underlying biological mechanisms (Mushtaq et al., 2024; Hannun et al., 2019). This makes BTS a reliable source of evidence for downstream learning tasks such as disease screening, treatment monitoring and emotion recognition (Ye et al., 2025; Poterucha et al., 2025). For example, in Alzheimer's disease, disease-related brain changes can disrupt neural synchronization and functional connectivity, leading to systematic alterations in EEG rhythms, so that recordings from patients and healthy controls tend to exhibit two distinguishable physiological patterns (Kopčanová et al., 2024; Chetty et al., 2024). BTS provide a key data foundation for clinical analysis, and therefore learning on BTS is an important problem.

Cross-subject generalization is a central challenge for deploying BTS classification models, because training is performed on recordings from a subset of subjects while evaluation targets unseen individuals (Rayatdoost et al., 2021;

---

[*]Corresponding author. [1]College of Computer Science, Nankai University, Tianjin, China [2]College of Cyber Science, Tianjin Key Laboratory of Interventional Brain-Computer Interface and Intelligent Rehabilitation, Key Lab of Data and Intelligent System Security, Frontiers Science Center for New Organic Matter, Nankai University, Tianjin, China. Correspondence to: Haoran Li <wanch_lhr@nankai.edu.cn>.

*Proceedings of the 43rd International Conference on Machine Learning*, Seoul, South Korea. PMLR 306, 2026. Copyright 2026 by the author(s).

Zhou et al., 2021b). As a result, performance often drops at test time for two reasons. First, inter-subject variability can mask class-relevant physiological patterns in raw time-series signals, making them harder to learn (Wang et al., 2024b; Roques et al., 2025). Even within the same class, recordings from different subjects may show large point-wise discrepancies, as illustrated in the top row of *Figure 1* (Sec. 5.3). Second, models may exploit subject-specific shortcuts, overfitting to individual idiosyncrasies rather than learning task-relevant physiological patterns (Yin et al., 2025; Kim et al., 2024) (Sec. 5.2).

Suppressing subject-specific variability is key to cross-subject generalization in BTS (Shen et al., 2023; Apicella et al., 2024). Prior work broadly follows two paradigms. First, stronger backbones, especially Transformer-based models, increase representation capacity via multi-scale temporal modeling and cross-channel interactions, which can implicitly reduce subject-specific variability (Wen et al., 2023). Second, adversarial methods explicitly encourage subject-invariant features by training a subject discriminator with gradient reversal layer (GRL) (Ganin et al., 2016) to penalize subject predictability while retaining label-relevant information (Zhou et al., 2024).

However, these paradigms largely treat subject-specific variability as an implicit nuisance and rarely ask a more fundamental question: **How should subject-specific variability in BTS signals be explicitly modeled?** Without an explicit characterization of what constitutes subject-specific variability, it remains unclear what the model should suppress, making invariance objectives difficult to control and potentially removing label-relevant structure.

To fill this gap, we introduce a new perspective for analyzing subject-specific variability through spectral structural patterns. Our motivation stems from a basic property of physiological signals: discriminative information is often expressed through task-relevant oscillatory bands or characteristic waveform structures rather than arbitrary feature dimensions. As shown in the second column of *Figure 1* (c) and (d), samples sharing the same label exhibit highly similar global spectral patterns despite substantial time-domain variation across subjects. Meanwhile, subject differences are often reflected as band-wise amplitude variation and phase shift over similar spectral structures. For instance, the red bars in *Figure 1* (c) reveal substantial energy disparities at 10 Hz across two subjects, which we attribute to individual differences. We term this phenomenon **spectral drift**. Additional examples are provided in Appendix A.

Based on this observation, we hypothesize that cross-subject variability can be approximated as label-conditioned spectral transformations, primarily manifested as magnitude scaling and phase rotation (Kim et al., 2024). This raises a natural question: can we align these transformations to suppress subject-specific variability while preserving task-relevant oscillatory structure?

Specifically, we introduce the Frequency-Band Alignment Module (FBAM), which models subject variability as structured spectral drift and performs adaptive frequency-band alignment in the Fourier domain. FBAM extracts compact band-wise statistics to characterize oscillatory structures and uses cross-attention between spectral descriptors and learnable band tokens to derive task-aware modulation coefficients for amplitude and phase adjustment. This design selectively enhances discriminative task-relevant frequency patterns while suppressing subject-dependent spectral variations. To further stabilize cross-subject representations, we propose Sample-Conditional Layer Normalization (SCLN), which adaptively calibrates sample-level feature statistics to mitigate residual distribution shifts without subject supervision. By integrating FBAM and SCLN into a Transformer-based backbone, we propose BioFormer for cross-subject biomedical time-series modeling. Experiments on six biomedical time-series benchmarks show that BioFormer consistently outperforms 12 strong baselines and multiple domain generalization methods, achieving up to 6% absolute F1-score improvement. In summary, the major contributions are as follows:

(1) We identify *spectral drift* as an important form of inter-subject variability and reformulate cross-subject generalization as a *frequency-domain structural alignment* problem, which we quantitatively validate to support the utility of this perspective.

(2) We propose BioFormer, a hybrid spectral–temporal Transformer that integrates FBAM with sample-adaptive normalization, and is explicitly designed to model and mitigate cross-subject variability in biomedical time-series data.

(3) We conduct comprehensive evaluations on 6 BTS benchmarks. BioFormer achieves rank-1 performance under cross-subject protocols, outperforming 12 baselines and 6 domain generalization methods. Code is available at `https://github.com/NKU-EmbeddedSystem/BioFormer`.

## 2. Related Work

**Transformers for Biomedical Time Series.** Transformer models have been widely adopted for BTS analysis due to their strong ability to capture long-range dependencies (Wen et al., 2023). In BTS applications, such models serve as powerful feature extractors that implicitly suppress subject-specific variability (Roy et al., 2019; Craik et al., 2019). Existing designs mainly emphasize multi-scale temporal modeling and cross-channel interactions in the time or latent feature space (Nie et al., 2023; Wang et al., 2024a; Liu et al., 2025), while cross-subject distribution drift is not

explicitly modeled. BioFormer departs from this paradigm by incorporating explicit spectral structure into the Transformer encoder, enabling direct and interpretable modeling of subject-dependent variability.

**Domain Generalization for Cross-Subject Task.** Cross-subject generalization can be viewed as a special case of domain generalization (DG), where each subject corresponds to a domain. Existing DG approaches typically improve robustness through adversarial invariance learning, feature-statistics alignment, or distribution perturbation (Ganin et al., 2016; Long et al., 2015; Sun & Saenko, 2016; Li et al., 2022). Recent BTS-specific DG methods further extend these strategies using subject-aware alignment and domain-invariant representation learning (Wang et al., 2024b; Tan et al., 2025; Huang et al., 2025). However, most existing methods primarily focus on enforcing alignment in the time domain or latent feature space, often suppressing subject discrepancy through generic feature matching objectives. In contrast, BioFormer explicitly models subject variability itself as structured spectral drift in the frequency domain. Rather than forcing feature-level invariance, Bio-Former adaptively models and calibrates band-wise spectral variation through frequency-aware modulation, enabling more effective preservation of task-relevant structures while reducing subject-dependent distortion. Additional comparisons between BioFormer and representative DG methods are provided in Appendix D.

**Frequency-Domain Modeling.** Frequency-domain analysis is fundamental to BTS research due to the inherently oscillatory nature of neural and cardiac signals (Pascucci et al., 2025). Accordingly, prior work has exploited spectral representations for feature extraction, noise suppression, and multi-view learning (Zhou et al., 2022; 2026). However, in most existing approaches, frequency information is treated in a largely static manner or utilized through global or heuristic transformations such as spectral normalization or adaptive filtering (Li et al., 2025). In practice, cross-subject differences often appear as structured, band-wise deviations in both magnitude and phase (Donoghue et al., 2020), which are largely decoupled from task semantics. We therefore treat frequency-domain structure as a first-class alignment target, enabling explicit, band-wise correction of subject-dependent spectral drift and directly supporting cross-subject generalization.

## 3. Problem Setup and Formulation

We study cross-subject classification for biomedical time-series data. Let $\mathcal{D} = \{(X_i, y_i, s_i)\}_{i=1}^{N}$ denote a dataset, where $X_i \in \mathbb{R}^{T \times C}$ is a multivariate time-series segment of length $T$ with $C$ channels, $y_i \in \mathcal{Y}$ is the corresponding class label, and $s_i \in \mathcal{S}$ denotes the subject identity. In the cross-subject setting, subjects are partitioned into disjoint training

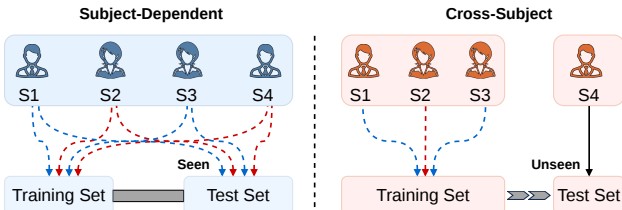

*Figure 2.* **Comparison between subject-dependent and cross-subject evaluation settings.** Left: subject-dependent evaluation, where samples from the same subjects may appear in both training and validation/test splits. Right: cross-subject evaluation, where training, validation, and test sets are partitioned by subject identity, and models are evaluated on completely unseen subjects without subject overlap across splits.

and test sets, denoted by $\mathcal{S}_{\text{train}}$ and $\mathcal{S}_{\text{test}}$, respectively, with $\mathcal{S}_{\text{train}} \cap \mathcal{S}_{\text{test}} = \emptyset$, as shown in *Figure 2*. A model $f_\theta$ is trained using samples from subjects in $\mathcal{S}_{\text{train}}$ and evaluated on samples from previously unseen subjects in $\mathcal{S}_{\text{test}}$.

## 4. Methodology

### 4.1. Overview

The overall architecture of BioFormer is shown in *Figure 3*. It comprises four sequential stages for cross-subject representation learning in biomedical time series. Given an input segment $X_i \in \mathbb{R}^{T \times C}$, BioFormer first applies an embedding layer, followed by Pyramid Convolutional Embedding (PCE) to generate multi-scale features $\{X_{i,j}\}$ with $X_{i,j} \in \mathbb{R}^{T_j \times D}$, where $T_j \in \{\frac{T}{2}, \frac{T}{4}, \frac{T}{8}\}$ via progressive convolutional downsampling (Details in Appendix B.1). The resulting multi-resolution features are fed in parallel into a 6-layer Hybrid Transformer Encoder, where each layer alternates between temporal self-attention and the Frequency-Band Alignment Module (FBAM) (Section 4.2). The three parallel encoder outputs are concatenated to form $h$, which is then computed by Sample Conditional Layer Normalization (SCLN) to obtain $h_{out}$ (Section 4.3). Finally, the resulting embedding is passed to a classifier based on multi-layer perceptron (MLP) to produce task predictions.

### 4.2. Frequency-Band Alignment Module

The Frequency-Band Alignment Module (FBAM) is the core component of BioFormer, designed to explicitly model and align subject-specific variability via band-wise dynamic modulation rather than individual frequency component. This design is motivated by two considerations: physiological time series exhibit semantically meaningful band structure (e.g., canonical EEG rhythms), and band-level aggregation improves robustness to local noise and minor frequency shifts.

Given a temporal feature representation $X_{i,j} \in \mathbb{R}^{T_j \times D}$, FBAM applies a one-dimensional discrete Fourier transform

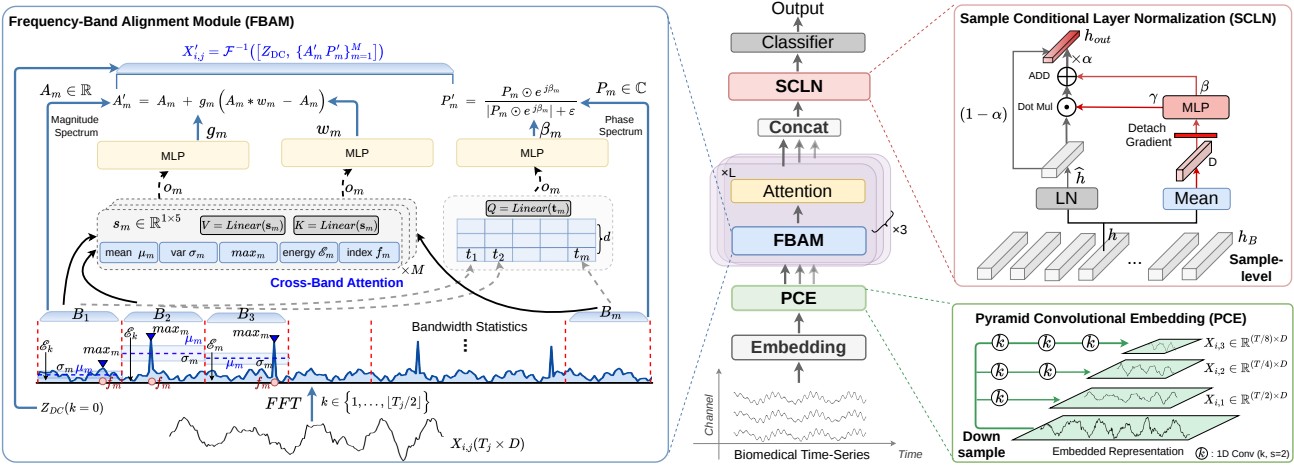

*Figure 3.* **Overall architecture of BioFormer.** The model consists of four main components: (1) a Pyramid Convolutional Embedding (PCE) module, (2) a Frequency-Band Alignment Module (FBAM), (3) a hybrid Transformer encoder that alternates FBAM and temporal self-attention layers, and (4) a Sample Conditional Layer Normalization (SCLN) module followed by a classification head.

(DFT) along the temporal axis for each feature channel:

$$Z_{i,j,k,d} = \mathcal{F}_r\big(X_{i,j}[:,d]\big)_k \in \mathbb{C}, \quad k = 0, \dots, \left\lfloor \frac{T_j}{2} \right\rfloor, \quad (1)$$

where $\mathcal{F}$ denotes the FFT operator, $k$ indexes the discrete frequency component, $d$ indexes the feature channel, and $Z_{i,j,k,d}$ is the complex Fourier coefficient.

Notably, the direct-current (DC) component ($k = 0$) mainly reflects the global signal baseline and offset. In FBAM, the DC component is kept fixed, and its magnitude and phase are added back only after modulating the non-DC frequency components. This is because the DC term primarily captures global amplitude bias rather than discriminative temporal dynamics. Allowing it to be adaptively modulated may cause the model to overfit subject- or dataset-specific baseline differences, thereby introducing nuisance bias into the spectral alignment process.

In biomedical time-series, task semantics are closely associated with structured oscillatory patterns in the spectral domain (Polich, 2007; Donoghue et al., 2020). Rather than independently modulating each frequency component, effective spectral alignment requires perceiving the band-level organization of physiological rhythms. FBAM addresses this by extracting statistical descriptors from predefined frequency bands, which serve as compact indicators of band-level physiological structure. These statistics enable the model to capture task-sensitive spectral patterns while mitigating subject-specific nuisance variations, thereby supporting structure-aware spectral modulation and improving cross-subject alignment. Specifically, the mean and standard deviation describe the spectral distribution, the peak magnitude and band energy characterize dominant oscillatory responses, and the dominant-frequency index captures the

frequency location of the strongest response within the band, reflecting which oscillatory frequency is most activated for the current sample.

Given the non-DC complex spectrum, we partition it into $M$ frequency bands $\{B_m\}_{m=1}^M$. For each band $B_m$, we compute a descriptor $\mathbf{s}_m \in \mathbb{R}^{1 \times 5}$:

$$\mathbf{s}_m = \big[ \log \mu_m, \ \log \sigma_m, \ \log max_m, \ \log \mathcal{E}_m, \ f_m \big], \quad (2)$$

where $\mu_m$ and $\sigma_m$ denote the mean and standard deviation of the magnitude spectrum, $max_m$ is the peak magnitude, $\mathcal{E}_m$ is the band energy, and $f_m$ is the dominant-frequency index within $B_m$. These statistics provide a low-dimensional yet informative summary for cross-subject comparison. Additional analysis and ablation studies on the choice of band-level statistics are provided in Appendix G.4.

FBAM maintains $M$ learnable band token embeddings $\{\mathbf{t}_m\}_{m=1}^M$, learned end-to-end. We project $\mathbf{s}_m$ and $\mathbf{t}_m$ to a shared latent space and fuse them via cross-attention:

$$\mathbf{o}_m = \text{CrossAttn}(\phi_o(\mathbf{t}_m), \phi_s(\mathbf{s}_m)) \in \mathbb{R}^{1 \times d}, \quad (3)$$

where $\phi_s$ and $\phi_o$ are linear projections. Finally, $\mathbf{o}_m$ is mapped to band-specific modulation parameters through three MLP heads,

$$w_m = \text{MLP}_w(o_m), g_m = \text{MLP}_g(o_m), \beta_m = \text{MLP}_p(o_m), \quad (4)$$

where $w_m$, $g_m$, and $\beta_m$ are modulation parameters produced from three independent MLP heads. $w_m$ and $g_m$ are used in Eq. 6 for magnitude modulation, whereas $\beta_m$ is used exclusively in Eq. 7 for phase modulation. Accordingly, $w_m$ acts as a dynamic convolution kernel for structure-aware band filtering, $g_m$ controls amplitude gain, and $\beta_m$ parameterizes phase rotation.

FBAM performs spectral alignment by explicitly modulating both magnitude and phase in each frequency band using the modulation factors. Importantly, FBAM does not rely on any explicit subject-variability objective or subject identity supervision. Instead, under task-level label supervision, the module learns to implicitly suppress subject-specific spectral variability by adaptively modulating band-wise magnitude and phase, thereby emphasizing label-relevant oscillatory structure while avoiding overly strong distribution alignment. A key design choice is to make alignment smooth and stable. Instead of directly overwriting the spectrum, we adopt a residual formulation so the module applies only the necessary correction, preserving label-relevant spectral structure whenever possible. This makes the modulation a controlled, band-wise refinement rather than an unconstrained transformation.

Following Eq. 1, we decompose the complex spectrum into amplitude and phase via the polar form:

$$Z_{i,j,k,d} = A_{i,j,k,d}\, e^{j\phi_{i,j,k,d}}, \qquad (5)$$

where $A_{i,j,k,d} = |Z_{i,j,k,d}| \geq 0$ is the amplitude and $\phi_{i,j,k,d} = \arg(Z_{i,j,k,d}) \in (-\pi, \pi]$ is the phase of the $k$-th frequency component.

**Magnitude alignment.** Given the modulation factors $(w_m, g_m)$ for band $B_m$, FBAM adjusts the magnitude spectrum in a residual manner:

$$A'_m = A_m + g_m\left(A_m * w_m - A_m\right), \qquad (6)$$

where $A_m = |X[m]|$, $*$ denotes one-dimensional convolution over the frequency axis. This residual update enables adaptive smoothing and redistribution of within-band energy while avoiding overly large perturbations.

**Phase alignment.** FBAM applies phase alignment on the unit-complex representation using the predicted phase coefficient $\beta_m$:

$$P'_m = \frac{P_m \odot e^{j\beta_m}}{|P_m \odot e^{j\beta_m}| + \varepsilon}, \qquad (7)$$

where $P_m = e^{j\phi[m]}$, $\odot$ is element-wise multiplication and $\varepsilon$ is a small constant for numerical stability. This operation implements a band-wise rotation in the complex plane, aligning phase drift while preserving band-wise phase representation.

Finally, the aligned magnitude and phase spectra are recombined with the preserved DC component and transformed back to the temporal domain via inverse Fourier transform $\mathcal{F}^{-1}$:

$$X'_{i,j} = \mathcal{F}^{-1}\left([Z_{\mathrm{DC}},\, \{A'_m P'_m\}_{m=1}^M]\right), \qquad (8)$$

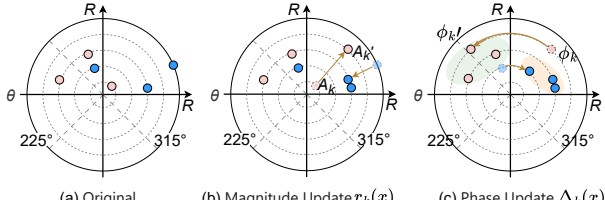

*Figure 4.* **Polar-view illustration of FBAM modulation in a Fourier subspace.** (a) The original coefficient in polar form. (b) Magnitude scaling $A'_k = r_k(x)A_k$ (radial change). (c) Phase rotation $\phi'_k = \phi_k + \Delta_k(x)$ (angular change). Together, $(r_k(x), \Delta_k(x))$ implement the scaling-and-rotation alignment within the $\{\cos, \sin\}$ subspace.

The resulting representation $X'_{i,j}$ aligns oscillatory structure while suppressing subject-specific variability. The FBAM procedure is provided in Appendix B.3.

**Interpretable Alignment in the Fourier subspaces.** A real-valued signal decomposes into independent 2D Fourier subspaces spanned by $\{\cos(2\pi kt/T), \sin(2\pi kt/T)\}$. For each frequency bin $k$, the complex coefficient $z_k = A_k e^{j\phi_k}$ is in one-to-one correspondence with a 2D real vector $\mathbf{a}_k = [a_k^{(c)}, a_k^{(s)}]^\top$. As shown in *Figure 4*, FBAM predicts a sample-conditioned complex modulator $u_k(x)$ for each bin $k$, parameterized as $u_k(x) = r_k(x)e^{j\Delta_k(x)}$, where $r_k(x) \geq 0$ controls magnitude scaling and $\Delta_k(x)$ induces phase rotation. A modulation $z'_k = u_k(x)\, z_k$ induces an exact *scaling + rotation* in that subspace:

$$\mathbf{a}'_k(x) = r_k(x)\, \mathbf{R}(\Delta_k(x))\, \mathbf{a}_k(x),$$
$$\mathbf{R}(\Delta) = \begin{bmatrix} \cos\Delta & -\sin\Delta \\ \sin\Delta & \cos\Delta \end{bmatrix}. \qquad (9)$$

Therefore, learning $(r_k(x), \Delta_k(x))$ provides an interpretable alignment mechanism within each Fourier subspace, rather than relying on static frequency-domain normalization (Appendix E). In FBAM, we aim to suppress subject-induced variability within each class while preserving inter-class separation. Under label supervision, the model can achieve this objective by adaptively modulating amplitude and phase in each Fourier subspace, effectively performing the scaling-and-rotation alignment illustrated in *Figure 4*(b) and (c). Further discussion is provided in Sec. 5.4 and Appendix F.

### 4.3. Sample Conditional Layer Normalization

While FBAM explicitly reduces subject-dependent spectral variation through band-wise alignment, overly strong alignment may also suppress informative sample-specific variability by driving same-label samples toward overly homogeneous representations. To mitigate this effect, we introduce Sample Conditional Layer Normalization (SCLN), which complements FBAM by performing sample-adaptive calibration at the encoder output. SCLN therefore intro-

duces a sample-adaptive residual normalization mechanism that preserves discriminative sample-level characteristics while stabilizing representation statistics across subjects.

Given an intermediate representation $\mathbf{h} \in \mathbb{R}^{T \times D}$, SCLN first applies standard layer normalization (LN), $\hat{\mathbf{h}} = \mathrm{LN}(\mathbf{h})$. Sample-adaptive scale and shift parameters are obtained by temporally averaging the input, blocking gradient propagation, and applying a lightweight MLP:

$$[\boldsymbol{\gamma}, \boldsymbol{\beta}] = \mathrm{MLP}(\mathrm{stopgrad}(\mathrm{Mean}_t(\mathbf{h}))), \quad (10)$$

where $\boldsymbol{\gamma}, \boldsymbol{\beta} \in \mathbb{R}^D$ are broadcast along the temporal dimension. The $\mathrm{stopgrad}(\cdot)$ operator prevents the normalization branch from directly reshaping the encoder representation during backpropagation, thereby stabilizing optimization and avoiding trivial feature adaptation.

Finally, adaptive normalization is applied via residual scale–shift blending:

$$\mathbf{h}_{\mathrm{out}} = (1 - \alpha)\hat{\mathbf{h}} + \alpha \left( \gamma \odot \hat{\mathbf{h}} + \beta \right), \quad (11)$$

where $\alpha$ is a fixed hyperparameter controlling the overall strength of sample-adaptive modulation (The discussion of $\alpha$ in Appendix G.7). This residual formulation allows SCLN to calibrate representation statistics without enforcing strict subject-invariant normalization. Consequently, BioFormer can suppress subject-dependent distributional drift while still preserving task-relevant sample diversity, which is particularly important under cross-subject evaluation.

# 5. Experiments

**Datasets** We evaluate BioFormer under a strict *cross-subject* protocol on six biomedical time-series benchmarks spanning both EEG and ECG modalities. These datasets cover two common labeling paradigms introduced in Sec. 3: (i) *subject-level labeling*, where all samples from a subject share the same class label (APAVA (Escudero et al., 2006), ADFTD (Miltiadous et al., 2023), PTB (Goldberger et al., 2000), PTB-XL (Wagner et al., 2020)), and (ii) *sample-level labeling*, where labels are assigned independently to each sample (BCI-2a (Blankertz et al., 2004) and BCI-2b (Blankertz et al., 2006)). Table 1 summarizes the dataset scale, task settings, and signal modalities.

**Baselines.** We compare BioFormer with Transformer-based time-series models, including the Vanilla Transformer (Vaswani et al., 2017) and recent variants such as Reformer (Kitaev et al., 2020), Informer (Zhou et al., 2021a), Autoformer (Wu et al., 2021), FEDformer (Zhou et al., 2022), Nonformer (Liu et al., 2022), Crossformer (Zhang & Yan, 2023), PatchTST (Nie et al., 2023), iTransformer (Liu et al., 2024), MTST (Zhang et al., 2024). We also include two strong baselines: Medformer (Wang et al., 2024a),

*Table 1.* **Summary of datasets and task settings.** "S-K" denotes a $K$-class *subject-level* classification setting, where all samples from the same subject share one label. "M-K" denotes a $K$-class *sample-level* classification setting, where each subject may contribute samples from multiple classes.

| Dataset | Subjects | Samples | Task | Channels | Modality |
|---------|----------|---------|------|----------|----------|
| APAVA | 23 | 5,967 | S-2 | 16 | EEG |
| ADFTD | 88 | 69,752 | S-3 | 19 | EEG |
| PTB | 198 | 64,356 | S-2 | 15 | ECG |
| PTB-XL | 17,596 | 191,400 | S-5 | 12 | ECG |
| BCI-2a | 9 | 5,184 | M-4 | 3 | EEG |
| BCI-2b | 9 | 6,520 | M-2 | 3 | EEG |

a SOTA medical Transformer, and SoftShape (Liu et al., 2025), a SOTA non-Transformer model.

**Implementation Details** All models are implemented in PyTorch. Models are trained using Adam with a learning rate of $1 \times 10^{-4}$ for up to 100 epochs on NVIDIA RTX 3090 GPUs, with early stopping if the validation F1-score does not improve for 10 consecutive epochs. Performance is measured using Accuracy, Recall, F1-score, AUROC, and AUPRC, averaged over five random seeds (41–45). In our implementation, the number of frequency bands $M$ is set to 6. Details on datasets, baselines, and implementation are provided in Appendix C.

## 5.1. Cross-Subject Generalization Performance

*Table 2* provides a comprehensive comparison against 12 baselines across six BTS datasets. We report $\mathrm{Avg. Rank}$, computed over six evaluation metrics (lower is better). Bio-Former achieves the best overall average rank and attains the top average performance on four of the six datasets, demonstrating strong and consistent generalization across diverse biomedical settings. Compared with the SOTA model Soft-Shape (Liu et al., 2025), BioFormer delivers clear gains on ADFTD and PTB, improving F1 by over 5%, and achieves higher AUROC (AUR.) on most datasets, indicating more reliable decision boundaries under cross-subject shifts. Computational complexity analysis is deferred to Appendix G.8.

## 5.2. Effectiveness of Alignment Paradigms

We study whether explicitly aligning structured spectral components is effective and stable for cross-subject task. Concretely, we compare FBAM with representative, plug-and-play DG methods (GRL (Ganin et al., 2016), MMD (Long et al., 2015), CORAL (Sun & Saenko, 2016), SubjectNorm (Rayatdoost et al., 2021), MixStyle (Zhou et al., 2021b), and DSU (Li et al., 2022)) that promote feature alignment through adversarial or statistical objectives. These baselines can be seamlessly integrated into a backbone via auxiliary losses or normalization-based augmentations. Implementation details are provided in Appendix C.4.

*Table 2.* Cross-subject generalization performance of different models across six datasets (complete results are reported in Appendix G.1). EEG-2 indicates the 2 classification task on EEG data. Best is highlighted with red background; second best with blue. Avg. Rank (A.Rank) is computed over 36 metrics (6 datasets × {Acc, Prec, Rec, F1, AUROC (AUR.), AUPRC}), lower is better. Win counts the number of datasets where a method achieves the best average score over the 6 metrics. † denotes results reproduced in our experimental environment, while all other baseline results are adopted from Medformer (Wang et al., 2024a).

| Models | APAVA (EEG-2) | | | ADFTD (EEG-3) | | | PTB (ECG-2) | | | PTB-XL (ECG-5) | | | BCI-2a (EEG-4) | | | BCI-2b (EEG-2) | | | A.Rank↓ | Win↑ |
|---|---|---|---|---|---|---|---|---|---|---|---|---|---|---|---|---|---|---|---|---|
| | Acc | F1 | AUR. | Acc | F1 | AUR. | Acc | F1 | AUR. | Acc | F1 | AUR. | Acc | F1 | AUR. | Acc | F1 | AUR. | | |
| Transformer [NeurIPS'17] | 76.30 | 73.75 | 72.50 | 50.47 | 48.09 | 67.93 | 77.37 | 68.47 | 90.08 | 70.59 | 59.05 | 88.21 | 41.68 | 41.17 | 68.88 | 70.86 | 70.83 | 77.82 | 7.29 | 0 |
| Reformer [ICLR'20] | 78.70 | 75.93 | 73.94 | 50.78 | 47.94 | 69.17 | 77.96 | 69.65 | 91.13 | 71.72 | 60.69 | 88.80 | 42.69 | 42.52 | 69.64 | 70.90 | 70.82 | 78.34 | 5.25 | 0 |
| Informer [AAAI'21] | 73.11 | 69.47 | 70.46 | 48.45 | 45.74 | 65.87 | 78.69 | 70.84 | 92.09 | 71.43 | 60.44 | 88.65 | 42.52 | 42.24 | 69.20 | 70.57 | 70.54 | 77.86 | 7.42 | 0 |
| Autoformer [NeurIPS'21] | 68.64 | 68.06 | 75.94 | 45.25 | 42.59 | 61.02 | 73.35 | 63.69 | 78.54 | 61.68 | 48.85 | 82.04 | 27.43 | 27.05 | 53.04 | 58.00 | 57.95 | 61.06 | 12.25 | 0 |
| FEDformer [ICML'22] | 74.94 | 73.51 | 83.72 | 46.30 | 43.91 | 62.62 | 76.05 | 67.14 | 85.93 | 57.20 | 47.89 | 82.13 | 29.06 | 28.72 | 54.58 | 60.99 | 60.82 | 66.19 | 10.67 | 0 |
| Nonformer [NeurIPS'22] | 71.89 | 69.74 | 70.55 | 49.95 | 46.96 | 66.23 | 78.66 | 70.90 | 89.37 | 70.56 | 59.10 | 88.32 | 42.67 | 42.51 | 69.31 | 70.71 | 70.60 | 78.79 | 7.12 | 0 |
| PatchTST [ICLR'23] | 67.03 | 55.97 | 65.65 | 44.37 | 41.97 | 60.08 | 74.74 | 64.36 | 88.79 | 73.23 | 62.61 | 89.74 | 40.64 | 40.13 | 67.63 | 73.26 | 73.17 | 81.69 | 8.17 | 1 |
| Crossformer [ICLR'23] | 73.77 | 68.93 | 72.39 | 50.45 | 45.50 | 66.48 | 77.75 | 68.55 | 73.30 | 62.59 | 90.02 | 42.48 | 41.61 | 69.22 | 70.58 | 70.39 | 77.54 | 6.47 | 0 |
| iTransformer [ICLR'24] | 74.55 | 72.30 | 85.59 | 52.60 | 46.79 | 67.26 | 83.89 | 79.06 | 91.18 | 69.28 | 56.20 | 86.71 | 33.58 | 33.23 | 59.57 | 65.94 | 65.84 | 72.17 | 7.94 | 0 |
| MTST [PMLR'24] | 71.14 | 64.01 | 68.87 | 45.60 | 44.31 | 62.50 | 76.59 | 67.38 | 86.86 | 72.14 | 61.43 | 88.97 | 39.31 | 39.06 | 66.61 | 71.53 | 71.50 | 79.48 | 8.33 | 0 |
| Medformer [NeurIPS'24] | 78.74 | 76.31 | 83.20 | 53.27 | 50.65 | 70.93 | 83.50 | 79.18 | 92.81 | 72.87 | 62.02 | 89.66 | 42.40 | 42.29 | 69.09 | 69.40 | 69.30 | 76.03 | 4.94 | 0 |
| SoftShape† [ICML'25] | 81.22 | 78.71 | 82.24 | 54.60 | 48.91 | 70.08 | 83.72 | 78.42 | 94.38 | 73.94 | 62.97 | 90.59 | 40.87 | 38.69 | 68.57 | 71.28 | 71.16 | 78.48 | 3.61 | 1 |
| **BioFormer† (Ours)** | 82.31 | 79.77 | 88.52 | 56.73 | 53.82 | 74.74 | 87.73 | 84.71 | 94.71 | 73.37 | 62.56 | 90.37 | 46.68 | 46.35 | 72.49 | 72.83 | 72.81 | 80.38 | 1.47 | 4 |

*Table 3.* Comparison of cross-subject generalization methods on three datasets. Accuracy and F1-score (%) are reported. Cell background color indicates the performance change relative to the base model without adaptation (None): red denotes strong improvement ($\Delta > 3$), blue denotes non-degrading or mild improvement ($(0 \leq \Delta \leq 3)$), and gray denotes degradation ($\Delta < 0$). The best result within each base model is highlighted in **bold**. The last two columns report stability metrics across all 16 settings: All baseline results in this table are reproduced under our unified experimental environment. Pos. Rate is the percentage of positive gains ($\Delta > 0$), and Worst $\Delta$ is the minimum change.

| Method | APAVA (EEG-2Class) | | | | | | ADFTD (EEG-3Class) | | | | | | PTB-XL (ECG-5Class) | | | | Stability | |
|---|---|---|---|---|---|---|---|---|---|---|---|---|---|---|---|---|---|---|
| | Trans. | | Medf. | | EEGNet | | Trans. | | Medf. | | EEGNet | | Trans. | | Medf. | | Pos. Rate (%) | Worst $\Delta$ |
| | Acc | F1 | Acc | F1 | Acc | F1 | Acc | F1 | Acc | F1 | Acc | F1 | Acc | F1 | Acc | F1 | | |
| None | 73.12 | 70.53 | 78.32 | 76.02 | 68.89 | 66.39 | 51.34 | 47.97 | 51.89 | 42.17 | 49.48 | 45.67 | 70.18 | 59.01 | 73.01 | 62.09 | – | – |
| +SubjNorm | 47.11 | 46.73 | 53.17 | 52.65 | 66.57 | 65.16 | 48.10 | 44.42 | 52.13 | 38.06 | 47.64 | 43.25 | 70.79 | 59.32 | 72.49 | 61.36 | 18.8 | -26.01 |
| +MMD | 75.82 | 71.44 | 75.14 | 72.44 | 71.88 | 67.95 | 44.29 | 20.46 | 40.07 | 24.16 | 48.05 | 36.95 | 63.39 | 46.26 | 73.06 | 57.29 | 31.2 | -27.51 |
| +CORAL | 78.57 | 75.70 | 76.03 | 73.74 | 69.13 | 66.26 | 50.70 | 47.08 | 49.80 | 40.39 | 49.41 | 41.36 | 70.83 | 58.67 | 73.02 | 62.11 | 37.5 | -4.31 |
| +GRL | 79.16 | 76.21 | 75.04 | 71.93 | 70.78 | 67.22 | 50.42 | 47.73 | 53.78 | 40.67 | 48.97 | 42.22 | 71.10 | 59.20 | 72.22 | 61.80 | 43.8 | -4.09 |
| +MixStyle | 73.91 | 71.43 | 76.39 | 73.94 | 62.87 | 61.72 | 50.36 | 47.05 | 51.76 | 42.35 | 51.15 | 46.24 | 70.50 | 59.18 | 72.93 | 61.96 | 43.8 | -6.02 |
| +DSU | 70.31 | 68.70 | 76.90 | 74.48 | 67.74 | 65.62 | 51.58 | 49.84 | 52.19 | 41.57 | 49.82 | 44.96 | 71.58 | 60.29 | 73.05 | 61.95 | 43.8 | -2.81 |
| **+FBAM (Ours)** | 81.61 | 80.00 | 83.03 | 81.46 | 75.42 | 73.89 | 54.38 | 52.13 | 51.86 | 47.08 | 52.82 | 49.13 | 71.26 | 60.90 | 73.01 | 61.80 | **81.2** | **-0.29** |

As shown in *Table 3*, **spectral alignment improves both performance and stability in cross-subject learning**. Red/blue cells denote improvements over the corresponding backbone-only setting, while gray cells indicate degradation. On APAVA with a Medformer backbone, FBAM improves F1 by over 10% compared with GRL, indicating substantially stronger cross-subject transfer. Notably, FBAM often improves the underlying backbones; for example, Medformer with FBAM reaches an F1 of 81.46 on APAVA, corresponding to a 5% improvement over the Medformer. To quantify stability, we report Pos. Rate, the fraction of settings with positive gains. BioFormer achieves 81%, outperforming the second-best method by nearly 40%, with a worst-case drop of only 0.3%. These results suggest that explicitly modeling structured spectral variability yields more reliable cross-subject alignment than generic feature-level discrepancy minimization.

### 5.3. Compared to Time-domain Alignment

To explore the benefit of aligning in the frequency domain, we implement a time-domain counterpart of FBAM, termed

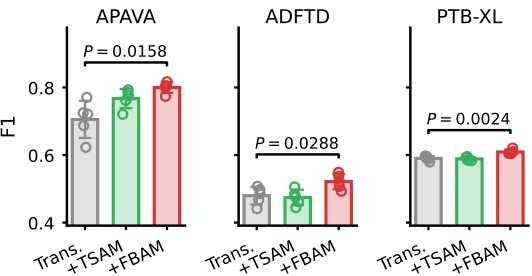

*Figure 5.* **Cross-subject evaluation of alignment domains.** F1 scores on three datasets. Dots denote individual runs. Welch's $t$-test; $P$ values shown.

*Temporal Segment Alignment Module (TSAM)*, more detail can be found in Appendix B.2. TSAM follows the same design philosophy and predicts modulation coefficients and adaptively recalibrates representations, but applies the modulation directly in the time domain. As shown in *Figure 5*, FBAM yields absolute F1-score improvements of 3.29% on APAVA (80.00 vs. 76.71), 4.68% on ADFTD (52.13 vs. 47.45), and 2.04% on PTB-XL (60.90 vs. 58.86). The

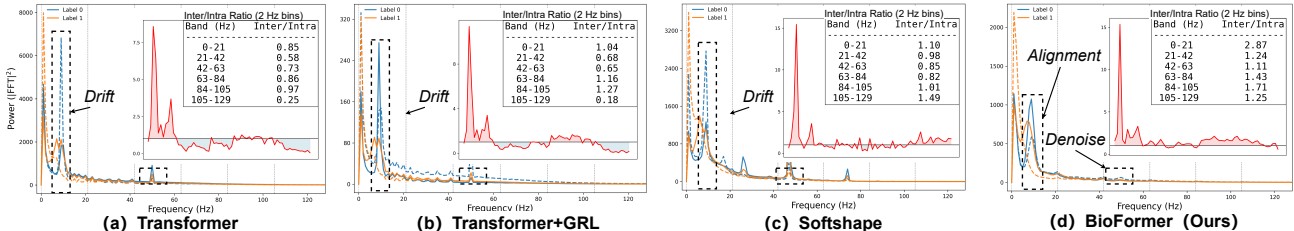

**(a) Transformer**     **(b) Transformer+GRL**     **(c) Softshape**     **(d) BioFormer (Ours)**

*Figure 6.* **A perspective of spectral structural alignment.** Power spectra of final-layer embeddings and the corresponding Frequency-Band Discriminability (FBD) distributions for different methods. The red curves show the FBD computed in 2 Hz frequency bins. A larger red area indicates *stronger spectral alignment* and *more prominent task-discriminative information*.

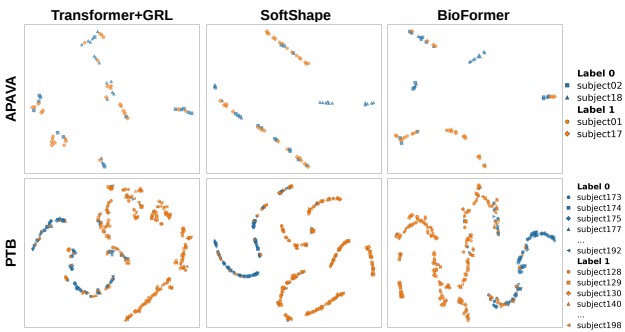

*Figure 7.* **t-SNE visualization of learned representations.** Colors indicate task labels and marker shapes denote subjects.

observed improvements are statistically significant, with all paired tests resulting in small $p$-values ($p < 0.02$). This result supports our hypothesis that subject-specific variability is obscured in the time domain and thus difficult to model, whereas spectral structural alignment enforces consistent oscillatory structure, thereby preserving semantics.

### 5.4. Spectral Structural Alignment Analysis

To further analyze whether BioFormer indeed aligns subject-dependent spectral structures, we introduce a metric termed *Frequency-Band Discriminability (FBD)*. For each frequency band $b$, we define the intra-class discrepancy $\text{Intra}(b)$ as the standard deviation of band-wise average spectral power across subjects within the same class, and the inter-class discrepancy $\text{Inter}(b)$ as the maximum difference in band-wise average spectral power between classes. Their ratio $\text{FBD} = \frac{\text{Inter}(b)}{\text{Intra}(b)}$ measures the degree to which task-related spectral structure dominates subject-specific variation, with $\text{FBD} > 1$ indicating favorable alignment. Detailed definitions are provided in Appendix F.

As shown in *Figure 6*, FBD is computed from the power spectra of the final-layer embeddings. The top-right subplot visualizes the FBD distribution over 2 Hz frequency bins, where red regions indicate $\text{FBD} > 1$ and blue regions indicate $\text{FBD} \leq 1$. BioFormer exhibits the largest red regions across most frequency bands, consistent with its strong cross-subject performance. In the low-frequency range (0–

*Table 4.* **Subject separability analysis.** Lower Subject F1, NMI, and AMI indicate weaker subject-specific information in the learned representation.

| Dataset | Metric | Trans.+GRL | SoftShape | BioFormer |
|---------|--------|-----------|-----------|-----------|
| APAVA | Subject F1 ↓ | 89.97 | 95.00 | **89.65** |
|  | NMI ↓ | 0.4563 | 0.6373 | **0.2787** |
|  | AMI ↓ | 0.4417 | 0.6179 | **0.2585** |
| PTB | Subject F1 ↓ | 99.06 | 98.33 | **97.95** |
|  | NMI ↓ | **0.4355** | 0.4701 | 0.4505 |
|  | AMI ↓ | 0.2808 | 0.2592 | **0.2260** |

20 Hz), which typically contains task-relevant semantics, BioFormer attains an FBD of 2.87, substantially higher than SoftShape (1.10). The regions highlighted by the black boxes further show that, within the same class, the spectra of two subjects become visibly closer under BioFormer. These observations suggest that, under task supervision, FBAM leverages magnitude scaling and phase rotation to align oscillatory components and suppress subject-induced variability while preserving class-discriminative structure.

### 5.5. Analysis of Subject Separability

We further analyze whether BioFormer suppresses subject-dependent information in the learned representations. Following a standard subject-probing protocol, we freeze the trained backbone and train a lightweight MLP probe to predict subject identities from the extracted features. Lower Subject F1 indicates that subject identity is less recoverable from the representation. As shown in Table 4, BioFormer achieves the lowest Subject F1 on both APAVA and PTB. On APAVA, BioFormer reduces Subject F1 from 95.00 (Soft-Shape) and 89.97 (Trans.+GRL) to 89.65. On PTB, Bio-Former further reduces Subject F1 to 97.95, compared with 98.33 for SoftShape and 99.06 for Trans.+GRL. These results suggest that subject identity becomes less recoverable under BioFormer, even without explicit subject supervision.

To further visualize the learned representations, we apply t-SNE to embeddings from unseen test subjects. As shown in Figure 7, samples are primarily organized by task labels rather than subject identities, while subjects remain intermixed within each semantic cluster, suggesting reduced

*Table 5.* **Ablation study of BioFormer with different module configurations**. F1-scores (%) are reported.

| FBAM | | SCLN | Datasets | | | |
|---|---|---|---|---|---|---|
| Mag. | Phase | | APAVA | ADFTD | PTB | PTB-XL |
| × | × | × | 74.23 | 44.81 | 81.38 | 61.45 |
| × | × | ✓ | 71.20 | 45.90 | 82.41 | 61.78 |
| ✓ | ✓ | × | 78.07 | 51.78 | 80.55 | **62.56** |
| ✓ | × | ✓ | 75.82 | 49.96 | 78.07 | 62.36 |
| × | ✓ | ✓ | 74.06 | 52.15 | **84.71** | 61.84 |
| ✓ | ✓ | ✓ | **79.77** | **53.82** | 83.06 | 62.53 |

subject dependency. Consistent with the visualization results, BioFormer also achieves substantially lower NMI and AMI on APAVA, reducing NMI from 0.6373 (SoftShape) and 0.4563 (Trans.+GRL) to 0.2787, and reducing AMI from 0.6179 and 0.4417 to 0.2585. On PTB, all methods already exhibit relatively weak subject clustering, while BioFormer still achieves the lowest AMI (0.2260), compared with 0.2592 for SoftShape and 0.2808 for Trans.+GRL. Additional silhouette-based analysis is provided in Appendix G.6.

## 5.6. Ablation Studies

*Table 5* summarizes the ablation results of BioFormer under different module configurations. Removing FBAM causes substantial performance degradation, with F1 drops of over 8% on both APAVA and ADFTD, demonstrating the importance of explicit spectral structural alignment for mitigating subject-specific variability. We further analyze the roles of amplitude and phase modulation. Using only magnitude modulation improves APAVA, while phase-only modulation performs best on PTB, which is consistent with the observation that PTB mainly exhibits phase inconsistency across subjects. Joint amplitude–phase modulation generally achieves the most stable overall performance across datasets. Removing SCLN leads to additional performance drops on APAVA, ADFTD, and PTB, indicating that sample-adaptive normalization further stabilizes cross-subject representations after spectral alignment. Overall, the full Bio-Former consistently achieves the best or near-best performance across datasets. Additional ablation analyses are provided in Appendix G.3 and G.4.

## 5.7. Visualization Analysis for FBAM

We visualize the intermediate feature representations before and after FBAM in *Figure 8*. In the figure, red indicates the magnitude of representation differences across temporal indices. After FBAM processing, the discrepancy between subjects sharing the same label is visibly reduced, suggesting that FBAM suppresses subject-dependent variation through band-wise spectral alignment. At the same time, the processed representations of different classes remain distinguishable, indicating that task-related semantic

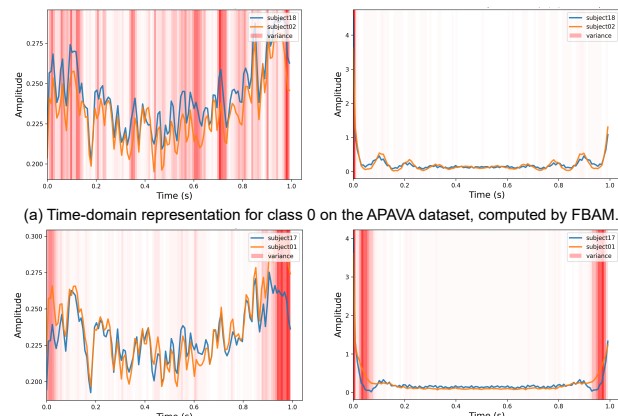

(a) Time-domain representation for class 0 on the APAVA dataset, computed by FBAM.

(b) Time-domain representation for class 1 on the APAVA dataset, computed by FBAM.

*Figure 8.* **Visualization of intermediate feature representations before and after FBAM alignment.** The right representation corresponds to the left after FBAM processing. Red indicates the magnitude of representation differences across temporal indices.

structure is preserved during alignment. These observations are consistent with the quantitative analyses in Sec. 5.5 and Appendix G.6, which show reduced subject separability while maintaining task-related clustering structure.

## 6. Limitations

BioFormer is most effective when cross-subject generalization is supported by shared task-relevant spectral structures, as commonly observed in physiological signals with stable oscillatory patterns. Its benefit may be reduced when such shared structures are weak or semantically uninformative in the frequency domain, for example in event-driven anomaly detection or irregular clinical time-series where labels are dominated by sparse transients or non-periodic local events.

## 7. Conclusion

We introduce *spectral drift* as a frequency-domain perspective for subject variability in biomedical time series and propose BioFormer for cross-subject generalization through adaptive frequency-band alignment. Experiments on six benchmarks show consistent gains over both general time-series models and domain generalization methods.

## Acknowledgments

This work was supported in part by Natural Science Foundation of China (62372254), Beijing-Tianjin-Hebei Basic Research Cooperation Project of Hebei Natural Science Foundation under Grant F2024203115(24JCZXJC00160 in Tianjin), Innovative Development Joint Fund Key Projects of Shandong NSF (ZR2023LZH003, ZR2022LZH009), The robotic AI-Scientist platform of Chinese Academy of Sciences.

## Impact Statement

This paper presents work whose goal is to advance the field of Machine Learning. There are many potential societal consequences of our work, none which we feel must be specifically highlighted here.

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

## Appendix Overview

This appendix provides additional analysis, implementation details, theoretical justification, and extended experimental results for BioFormer.

Appendix A presents a motivation study on the PTB dataset, showing that cross-subject variability is prominent in the time domain and is largely caused by phase inconsistency, which motivates modeling subject effects as structured spectral drift.

Appendix B details the architecture of BioFormer, including the Pyramid Convolutional Embedding (PCE), the purely time-domain baseline TSAM for controlled comparison, and the complete FBAM forward computation with its corresponding algorithm.

Appendix C describes the experimental setup, including dataset preprocessing, subject-level splitting protocols, baseline implementations, and training details for BioFormer and domain generalization baselines.

Appendix D further compares BioFormer with representative domain generalization methods and discusses the differences between spectral alignment and conventional discrepancy-minimization strategies.

Appendix E provides a theoretical analysis showing that amplitude and phase modulation correspond to scaling and rotation within Fourier subspaces, supporting the interpretability of frequency-domain alignment.

Appendix F introduces the Frequency-Band Discriminability (FBD) metric for analyzing cross-subject spectral separability and provides a corollary showing that reducing intra-class spectral drift improves discriminability.

Appendix G reports complete experimental results and additional analyses that complement the main paper. Specifically, Appendix G.1 presents the full cross-subject benchmark comparisons across all datasets and evaluation metrics. Appendix G.2 provides qualitative visualization of FBAM, demonstrating how frequency-band alignment reduces inter-subject variation in intermediate representations. Appendix G.3 presents ablation studies on FBAM, including analyses of band partitioning, DC modeling, residual magnitude correction, static band modulation, and attention design. Appendix G.4 further studies the effect of different band-level statistical descriptors for constructing frequency descriptors. Appendix G.5 compares BioFormer with BTS-specific domain generalization methods. Appendix G.6 analyzes subject-specific variability using silhouette scores to better understand representation behavior. Appendix G.7 studies the sensitivity of the SCLN mixing factor $\alpha$, while Appendix G.8 analyzes the trade-off between performance and computational complexity. Finally, Appendix G.9 reports subject-dependent evaluation results to complement the primary cross-subject setting and further examine the robustness of BioFormer under reduced distribution shift.

Overall, these analyses provide a comprehensive view of BioFormer from the perspectives of effectiveness, interpretability, robustness, representation behavior, and computational efficiency.

## A. Motivation

As illustrated in Figure 9, cross-subject variability in the PTB dataset is pronounced in the time domain and is largely driven by phase inconsistency in ECG waveforms. Even under the same diagnostic label, signals from different subjects exhibit noticeable misalignment at specific temporal locations, resulting in high-variance regions that are not directly related to label semantics. This temporal misalignment makes it easy for time-domain models to capture subject-dependent waveform idiosyncrasies rather than disease-relevant patterns. In contrast, the frequency-domain representation reveals a more structured and stable view: samples sharing the same label preserve highly similar global spectral envelopes, while inter-subject differences are mainly reflected as localized phase shifts (and to a lesser extent, magnitude variations) within specific frequency components. This observation suggests that, for PTB, cross-subject variability can be effectively characterized as structured spectral drift dominated by phase inconsistency.

## B. Module Design

### B.1. Pyramid Convolutional Embedding (PCE)

The **Pyramid Convolutional Embedding (PCE)** module acts as a hierarchical temporal feature extractor in BioFormer, producing multi-scale representations from biomedical time-series signals. Given an input segment $X_i \in \mathbb{R}^{T \times C}$, where $T$ denotes the sequence length and $C$ the number of channels, PCE first maps the input into a latent space via a token

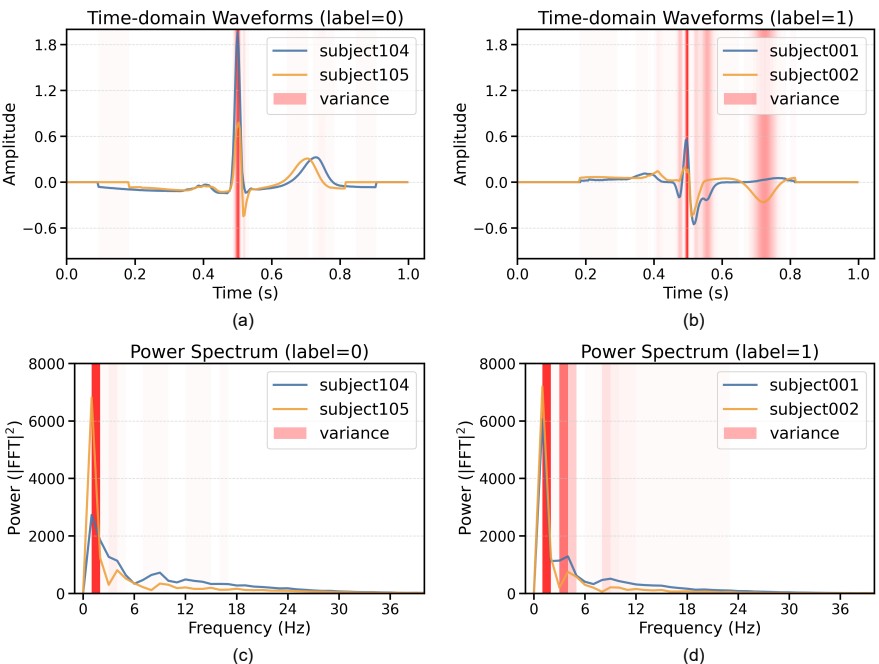

*Figure 9.* **Time and frequency domain analysis on the PTB dataset.** The first row ((a), (b)) shows time-domain waveforms, and the second row ((c), (d)) shows the corresponding power spectra. Colored curves denote signals from different subjects under the same label, while the red background indicates inter-subject variance, with darker regions representing larger discrepancies. As observed, cross-subject variability in PTB is primarily reflected as phase inconsistency in the time domain, which leads to pronounced variance despite similar spectral magnitude distributions.

embedding combined with positional encoding:

$$E_i = \text{TokenEmbed}(X_i) + \text{PosEmbed}(X_i).$$

The embedded representation $E_i \in \mathbb{R}^{T \times D}$ is subsequently processed by a cascade of one-dimensional convolutional downsampling stages with increasing stride factors $\{2, 4, 8\}$. Each stage comprises stacked Conv–BN–GELU blocks that progressively reduce the temporal resolution while keeping the embedding dimension $D$ fixed, resulting in a pyramid of temporal feature sequences:

$$\{X_{i,1}, X_{i,2}, X_{i,3}\} = \{f_1(E_i), f_2(E_i), f_3(E_i)\}, \quad X_{i,j} \in \mathbb{R}^{T_j \times D},$$

where $T_j \in \{T/2, T/4, T/8\}$ corresponds to the temporal scale at level $j$. This multi-resolution structure enables PCE to jointly model short-term oscillatory patterns and longer-range temporal dynamics in physiological signals.

To enhance cross-subject robustness and generalization, stochastic perturbations are incorporated into each convolutional stage through a pool of lightweight temporal augmentations. During each forward pass, a single augmentation (e.g., additive noise, random scaling, or dropout) is randomly sampled and applied to the intermediate feature representation:

$$\tilde{X}_{i,j} = Aug_r(X_{i,j}),$$

where $Aug_r$ denotes the selected augmentation operator (Jitter, Scale, Dropout, or Identity). Such perturbations regularize the learned features against channel-level variations and subtle inter-subject discrepancies commonly observed in EEG and ECG data.

Finally, PCE produces a set of multi-scale embeddings $\{X_{i,1}, X_{i,2}, X_{i,3}\}$, which serve as the hierarchical inputs to subsequent frequency-band alignment and attention encoding modules. By preserving channel dimensionality while progressively reducing temporal length, PCE achieves a favorable trade-off between representational capacity and computational efficiency.

### B.2. Temporal Segment Alignment Module (TSAM)

The **Temporal Segment Alignment Module (TSAM)** is a purely time-domain alignment module introduced as a controlled baseline to contrast with FBAM. Unlike frequency-aware alignment, TSAM corrects subject-specific distribution drift by

operating on uniformly segmented temporal structures, without invoking explicit frequency decomposition.

Given an input sequence $X_i \in \mathbb{R}^{T \times C}$, TSAM partitions the temporal axis into $M$ non-overlapping segments of approximately equal length:

$$X_i \xrightarrow{\text{Uniform Temporal Split}} \{X_{i,m}\}_{m=1}^{M},$$

where each segment $X_{i,m} \in \mathbb{R}^{T_m \times C}$ corresponds to a contiguous temporal interval and $\sum_{m=1}^{M} T_{i,m} = T_i$. This segmentation preserves the original signal ordering while providing a coarse temporal structure for segment-wise analysis.

For each temporal segment, TSAM computes a set of low-order statistical descriptors that summarize its global characteristics, including magnitude, variability, peak response, and energy. These segment-level descriptors are projected into a latent embedding space and aggregated through a cross-segment attention mechanism, which produces segment-specific control representations.

Conditioned on the learned segment representations, TSAM applies dynamic temporal convolution and residual gain modulation independently within each segment. The resulting segment-wise adjustments are restricted to local temporal neighborhoods defined by the segment boundaries. Finally, the updated segments are concatenated in their original temporal order to reconstruct the output sequence:

$$X_{i \, \text{out}} = \text{Concat}\big(\tilde{X}_{i,1}, \tilde{X}_{i,2}, \ldots, \tilde{X}_{i,M}\big).$$

All operations in TSAM are confined to the time domain and are defined with respect to explicit temporal segments rather than frequency-oriented structures. By design, TSAM isolates the effect of temporal partitioning and segment-wise modulation, enabling a direct comparison with frequency-based alignment mechanisms. For a fair and controlled comparison, TSAM adopts the same architectural hyperparameters as FBAM, including the convolution kernel size, control-token embedding dimension, and the number of segments/bands, differing only in whether alignment is performed over temporal segments or frequency bands.

### B.3. FBAM forward pass

Algorithm 1 summarizes the forward computation of FBAM. Given temporal features $X \in \mathbb{R}^{B \times T \times D}$, we first apply a real-valued FFT along the temporal axis and split the spectrum into the DC component and the remaining oscillatory coefficients. The DC term is preserved to avoid injecting subject-specific baseline bias, while the non-DC spectrum is converted into magnitude $A$ and unit-complex phase representation $P$. We then partition the non-DC frequencies into $K$ contiguous bands $\{B_k\}_{k=1}^{K}$ and compute band-wise statistical descriptors (e.g., moments, extrema, and energy), which are stacked and projected to obtain compact band context embeddings. Next, we perform a cross-band interaction between learned band tokens and the projected statistics to produce band-aware representations, from which FBAM predicts three sets of modulation parameters: a frequency kernel $W$ and gain $g$ for magnitude correction, and a phase offset $\beta$ for phase adjustment. For each band, magnitude alignment is implemented via a residual convolutional update $A \leftarrow A + g \cdot (A * W - A)$, while phase alignment is performed by applying a multiplicative unit-complex update and re-normalizing to maintain unit magnitude. Finally, the adapted magnitude and phase are recombined with the preserved DC component to form the aligned spectrum, and an inverse real FFT transforms the result back to the temporal domain, yielding the frequency-adapted features $X'$.

## C. Experimental Setup

### C.1. Dataset Preprocessing

In this appendix, we provide detailed descriptions of the datasets and preprocessing procedures used in our experiments. All datasets are evaluated under a unified cross-subject setting, where subjects are split at the identity level into training, validation, and test sets, and all samples from the same subject are assigned to the same split. This setting strictly prevents subject leakage and ensures a fair evaluation of cross-subject generalization.

The datasets include both subject–label aligned disease diagnosis benchmarks and subject–label non-aligned motor imagery benchmarks. The former category covers APAVA, ADFTD, PTB, and PTB-XL, where each subject is associated with a single diagnostic label. The latter category includes BCI-2a and BCI-2b, where each subject contributes samples from multiple classes.

---

**Algorithm 1** FBAM Forward Pass

---

**Require:** Temporal features $X \in \mathbb{R}^{B \times T \times D}$; number of bands $M$; band tokens $\mathbf{T} \in \mathbb{R}^{M \times d}$; kernel size $r$.
**Ensure:** Frequency-adapted features $X' \in \mathbb{R}^{B \times T \times D}$.
1: **I. FFT and DC split.**
2: $\hat{Z} \leftarrow \text{rFFT}(X_{(B,D,T)}^{\top}); \hat{Z}_{\text{DC}} \leftarrow \hat{Z}[:,:,0], \hat{Z}_{\text{rest}} \leftarrow \hat{Z}[:,:,1:].$
3: $A \leftarrow |\hat{Z}_{\text{rest}}|; P \leftarrow \hat{Z}_{\text{rest}}/(A + \varepsilon).$
4: Preserve $\hat{Z}_{\text{DC}}$ and adapt only oscillatory components.
5: **II. Frequency-band partitioning and statistics.**
6: Partition frequency indices into $M$ bands $\{B_m\}_{m=1}^{M}$.
7: **for** $m = 1$ **to** $M$ **do**
8:     $A_m \leftarrow A[:,:,B_m]; \mathbf{s}_m \leftarrow [\log \mu_m, \log \sigma_m, \log \max_m, \log \mathcal{E}_m, f_m].$
9: **end for**
10: Stack $\mathbf{S} \in \mathbb{R}^{B \times M \times 5}$ and project $\mathbf{C} \leftarrow \text{Proj}(\mathbf{S}).$
11: **III. Cross-band interaction.**
12: $\mathbf{O} \leftarrow \text{Interact}(\mathbf{T}, \mathbf{C}) \in \mathbb{R}^{B \times M \times d}.$
13: **IV. Band-wise modulation parameters.**
14: $W \leftarrow \text{Softmax}(\text{MLP}_w(\mathbf{O})), g \leftarrow \tanh(\text{MLP}_g(\mathbf{O})), \beta \leftarrow \tanh(\text{MLP}_\phi(\mathbf{O})).$
15: **V. Band-wise magnitude and phase alignment.**
16: **for** $m = 1$ **to** $M$ **do**
17:     **Magnitude:** $\Delta A_m \leftarrow A[:,:,B_m] * W_{:,m,:} - A[:,:,B_m].$
18:     $A[:,:,B_m] \leftarrow A[:,:,B_m] + g_{:,m} \cdot \Delta A_m.$
19:     **Phase:** $\Delta P_m \leftarrow \exp(j\, \beta_{:,m}).$
20:     $\tilde{P}_m \leftarrow P[:,:,B_m] \odot \Delta P_m.$
21:     $P[:,:,B_m] \leftarrow \tilde{P}_m/(|\tilde{P}_m| + \varepsilon).$
22: **end for**
23: **VI. Inverse FFT reconstruction.**
24: $\hat{Z}' \leftarrow [\hat{Z}_{\text{DC}}, A \odot P]; X' \leftarrow \text{irFFT}(\hat{Z}')_{(B,T,D)}^{\top}.$
25: **return** $X'$

---

**APAVA** (Escudero et al., 2006) is a public EEG dataset collected by the Alzheimer's Patients' Relatives Association of Valladolid, containing recordings from 23 subjects (12 Alzheimer's disease patients and 11 healthy controls). Each subject provides multiple 5-second trials recorded with 16 channels at a sampling rate of 256 Hz, resulting in 1,280 timestamps per trial. Each trial is first normalized using a standard scaler and then segmented into nine half-overlapping 1-second samples, yielding 5,967 samples in total. Each sample is associated with its originating subject ID. We follow a fixed subject-independent split: subjects with IDs $\{15, 16, 19, 20\}$ are used for validation, $\{1, 2, 17, 18\}$ for testing, and the remaining subjects for training.

**ADFTD** (Miltiadous et al., 2023) is an EEG dataset consisting of recordings from 88 subjects across three diagnostic categories: Alzheimer's disease (AD), Frontotemporal Dementia (FTD), and healthy controls. Signals are recorded with 19 channels at a raw sampling rate of 500 Hz. Each subject contributes one long continuous trial, with an average duration of approximately 12–14 minutes. A band-pass filter of 0.5–45 Hz is applied, after which signals are downsampled to 256 Hz and segmented into non-overlapping 1-second windows, discarding segments shorter than 1 second. This process yields 69,752 samples in total. For subject-independent evaluation, subjects are randomly split into training, validation, and test sets with a 60/20/20 ratio, and all samples from the same subject are assigned to the same split.

**PTB** (Goldberger et al., 2000) is a public ECG dataset originally collected from 290 subjects with 15 channels and multiple cardiac conditions. Following common practice, we use a subset of 198 subjects including myocardial infarction patients and healthy controls. Signals are downsampled from 1,000 Hz to 250 Hz and normalized using standard scalers. ECG recordings are further processed into individual heartbeats via R-peak detection across channels, with outlier beats removed. Each heartbeat is extracted around the R-peak and zero-padded to a fixed length based on the maximum duration, yielding 64,356 heartbeat samples. Subjects are split into training, validation, and test sets with a 60/20/20 ratio under the subject-independent setting.

**PTB-XL** (Wagner et al., 2020) is a large-scale ECG dataset containing recordings from 18,869 subjects with 12 channels and five diagnostic categories. To ensure label consistency, subjects with conflicting diagnoses across trials are removed, resulting in 17,596 subjects. We use the 500 Hz version of the recordings, which are downsampled to 250 Hz and normalized. Each 10-second trial is segmented into non-overlapping 1-second windows, discarding segments shorter than 1 second, yielding 191,400 samples. We allocate 60%, 20%, and 20% of subjects to the training, validation, and test sets, respectively.

**BCI Competition IV 2a (BCI-2a)** (Blankertz et al., 2004) is a four-class motor imagery dataset (left hand, right hand, tongue, and foot), containing EEG recordings from 9 subjects. Signals are originally recorded at 250 Hz with 22 EEG channels. Following common practice in motor imagery analysis, we retain only 3 bipolar channels (C3, Cz, and C4) that are most relevant to MI tasks. Each subject participates in two sessions with 288 trials per session. We merge samples at the subject level to construct a subject-independent evaluation protocol. To reduce temporal redundancy, signals are downsampled by a factor of 4. Subjects $\{6, 8\}$ are used for validation, $\{7, 9\}$ for testing, and the remaining subjects for training, with all samples from the same subject assigned to the same split.

**BCI Competition IV 2b (BCI-2b)** (Blankertz et al., 2006) is a binary motor imagery dataset involving left-hand and right-hand tasks. The dataset includes 9 subjects, each with 5 sessions recorded at 250 Hz using 3 bipolar channels (C3, Cz, and C4). The first two sessions contain data without feedback, while the remaining three include visual feedback. Samples are merged at the subject level and downsampled. We follow the same subject-independent split as in BCI-2a, using subjects $\{6, 8\}$ for validation, $\{7, 9\}$ for testing, and the rest for training.

## C.2. Baselines

**Transformer** (Vaswani et al., 2017) is the vanilla attention-based sequence model that serves as the fundamental backbone for subsequent Transformer variants.

**Reformer** (Kitaev et al., 2020) improves the efficiency of self-attention by introducing locality-sensitive hashing, enabling scalable modeling of long sequences.

**Informer** (Zhou et al., 2021a) employs probabilistic sparse attention and a distillation mechanism to reduce computational complexity in long time-series forecasting.

**Autoformer** (Wu et al., 2021) introduces a series decomposition architecture that explicitly separates trend and seasonal components for time-series modeling.

**FEDformer** (Zhou et al., 2022) performs attention computation in the frequency domain using Fourier or wavelet bases to improve efficiency and representation capacity.

**Nonformer** (Liu et al., 2022) replaces self-attention with a non-attentive mechanism based on global aggregation to reduce redundancy in sequence modeling.

**Crossformer** (Zhang & Yan, 2023) models long-term dependencies via cross-scale attention by hierarchically aggregating temporal segments.

**PatchTST** (Nie et al., 2023) segments time series into patches and applies Transformer modeling over patch-level representations to enhance local pattern modeling.

**iTransformer** (Liu et al., 2024) inverts the temporal and variate dimensions, treating each variable as a token to better capture inter-variable dependencies.

**MTST** (Zhang et al., 2024) integrates multi-scale temporal modeling with Transformer architectures to capture hierarchical temporal structures.

**Medformer** (Wang et al., 2024a) is a biomedical Transformer that incorporates multi-granularity temporal modeling tailored for physiological time-series signals.

**SoftShape** (Liu et al., 2025) is a non-Transformer time-series model that learns shape-based representations and employs a mixture-of-experts architecture for efficient and robust classification.

## C.3. Implementation Details

All experiments are implemented in PyTorch and conducted under fixed training, validation, and test splits. We evaluate all models using six macro-averaged metrics: Accuracy, Precision, Recall, F1-score, AUROC, and AUPRC. For all Transformer-based models (including BioFormer and baselines), we adopt a unified configuration with 6 encoder layers, model dimension $D = 128$, and feed-forward hidden dimension $D_{\text{ff}} = 256$. Models are optimized using Adam with a learning rate of $1 \times 10^{-4}$ and trained for up to 100 epochs, with early stopping triggered if the validation F1-score does not improve for 10 consecutive epochs. BioFormer, including FBAM and SCLN, is trained end-to-end using only the task classification objective without any auxiliary subject-alignment loss or subject identity supervision. The model achieving the best

validation performance is selected for final testing. Batch sizes are set to $\{32, 128, 128, 128, 128, 128\}$ for APAVA, ADFTD, PTB, PTB-XL, BCI-2a, and BCI-2b, respectively. Each experiment is repeated with five fixed random seeds (41–45), and we report the mean and standard deviation of all metrics. For SoftShape, we adopt the recommended configuration: two shape blocks, an MoE auxiliary-loss weight of 0.01, and a five-epoch linear warm-up schedule. All experiments are executed on 8 NVIDIA GeForce RTX 3090 GPUs.

For **BioFormer**, different group convolution strategies are adopted in FBAM for spectral magnitude modulation depending on the task setting. For multi-class classification tasks ($K > 2$), we use *batch-shared band-wise group convolution*, where convolution kernels are averaged across samples and shared within each frequency band to promote stable and globally consistent spectral modulation. For binary classification tasks, we instead apply *sample-specific band-wise group convolution* to enable finer-grained adaptation to individual spectral variations. Unless otherwise specified, FBAM employs a convolution kernel size of 3, a band-token embedding dimension of 64, and partitions the non-DC spectrum into 6 uniformly spaced frequency bands. For phase alignment, different strategies are applied across datasets: as illustrated in Figure 9, cross-subject variability in the PTB dataset is predominantly driven by phase inconsistency in ECG waveforms, and we therefore apply phase-only adjustment; for all other datasets, magnitude and phase are jointly modulated to address more general spectral drift. Sample Conditional Layer Normalization (SCLN) is applied to all datasets except PTB-XL, as the large scale and high diversity of PTB-XL render standard normalization sufficiently stable, while additional sample-adaptive modulation does not yield consistent gains. For the remaining datasets, SCLN effectively mitigates residual sample-level distribution shifts, with the residual blending coefficient $\alpha$ set to 0.1 for APAVA, ADFTD, BCI-2a, and BCI-2b, 0.2 for PTB, and 0 for PTB-XL. Data augmentation is performed by randomly selecting from `none`, `jitter`, and `scale`, where the numeric suffix indicates the augmentation intensity. The adopted augmentation schemes for APAVA, ADFTD, PTB, PTB-XL, BCI-2a, and BCI-2b are $\{$`scale0.1, drop0.25`$\}$, $\{$`drop0.5`$\}$, $\{$`none, drop0.5`$\}$, $\{$`jitter0.2, scale0.2, drop0.5`$\}$, $\{$`scale0.1, drop0.25`$\}$, and $\{$`scale0.1, drop0.25`$\}$, respectively.

### C.4. Domain Generalization Baselines and Implementation Details

This appendix details the implementation of domain generalization (DG) baselines evaluated in Section 5.2. All methods are evaluated on three backbones—Transformer, Medformer, and EEGNet (Lawhern et al., 2018)—across three datasets (APAVA, ADFTD, and PTB-XL), using identical cross-subject splits and optimization settings.

**SubjectNorm.** SubjectNorm is applied as a preprocessing step during data loading. For each subject, we perform subject-wise $z$-score normalization by computing the mean and standard deviation over all time points and channels belonging to that subject, and normalizing the corresponding samples accordingly. This operation is performed once before training and does not introduce additional loss terms.

**MMD.** MMD is implemented as an auxiliary alignment loss applied to intermediate feature representations. Features are extracted by temporally averaging the encoder outputs. During training, if a mini-batch contains samples from at least two subjects, two subjects are randomly selected and the MMD loss is computed between their feature distributions. The total training loss is

$$\mathcal{L} = \mathcal{L}_{\text{cls}} + 0.05 \cdot \mathcal{L}_{\text{MMD}}.$$

If a batch contains samples from only one subject, the MMD loss is skipped.

**CORAL.** CORAL follows the same training protocol as MMD, but aligns second-order feature statistics. The CORAL loss is computed between covariance matrices of features from two randomly selected subjects within a mini-batch. The final objective is

$$\mathcal{L} = \mathcal{L}_{\text{cls}} + 0.05 \cdot \mathcal{L}_{\text{CORAL}}.$$

As with MMD, the alignment loss is omitted when only one subject appears in the batch.

**GRL.** For adversarial alignment, we attach a subject discriminator to the encoder output via a gradient reversal layer (GRL). The discriminator predicts subject identities, while gradients flowing into the encoder are reversed. The reversal coefficient follows the standard DANN schedule and increases progressively during training. The training objective is

$$\mathcal{L} = \mathcal{L}_{\text{cls}} + 0.1 \cdot \mathcal{L}_{\text{subj}},$$

where $\mathcal{L}_{\text{subj}}$ denotes the subject classification loss.

**MixStyle.** MixStyle is implemented as a stochastic feature-level augmentation that mixes channel-wise mean and variance statistics across samples within a mini-batch. For Transformer and Medformer backbones, MixStyle is applied immediately

after the embedding layer, while for EEGNet it is inserted after the first convolutional block. MixStyle does not introduce an additional loss term and is active only during training.

**DSU.** DSU perturbs feature statistics by injecting learnable uncertainty into channel-wise mean and variance. For Transformer-based backbones, DSU is applied to embedding features, and for EEGNet it is inserted after the first convolutional block. Similar to MixStyle, DSU operates only during training and does not modify the loss function.

**FBAM.** FBAM is integrated into the encoder as a frequency-domain alignment module. FreqBandDrift blocks are inserted between successive encoder layers and operate directly on temporal feature sequences. FBAM modulates spectral magnitude and phase in a band-wise manner and is trained end-to-end using only the task classification loss, without introducing auxiliary alignment losses.

## D. Comparison with Domain Generalization Methods

This section further contextualizes BioFormer with respect to existing domain generalization (DG) methods for cross-subject biomedical time-series (BTS) analysis.

Most DG methods reduce subject discrepancy through adversarial alignment, statistical matching, or feature perturbation (Ganin et al., 2016; Long et al., 2015; Sun & Saenko, 2016; Zhou et al., 2021b; Li et al., 2022). Recent BTS-specific approaches, such as DMMR (Wang et al., 2024b), SEDA-EEG (Tan et al., 2025), and CADAN (Huang et al., 2025), further introduce subject-aware alignment mechanisms. However, these methods mainly operate in the time domain or latent feature space and generally do not explicitly model the spectral structure of subject variability.

In contrast, BioFormer is motivated by the observation that cross-subject variability in BTS often manifests as structured spectral drift. Rather than relying on explicit discrepancy minimization, BioFormer directly models and aligns band-wise spectral variation through frequency-aware modulation.

Table 6 compares representative Transformer backbones from the perspective of BTS modeling properties. Most existing architectures mainly improve temporal representation learning, while BioFormer additionally introduces explicit frequency-structured modeling tailored for cross-subject BTS analysis.

*Table 6.* Comparison of representative Transformer backbones for BTS modeling.

| Model | Multi-scale | Multi-granularity | Frequency-aware Structure | Designed for BTS |
|---|:---:|:---:|:---:|:---:|
| Transformer (Vaswani et al., 2017) / Reformer (Kitaev et al., 2020) / Informer (Zhou et al., 2021a) | × | × | × | × |
| Autoformer (Wu et al., 2021) / FEDformer (Zhou et al., 2022) | ✓ | × | ✓ | × |
| Crossformer (Zhang & Yan, 2023) / PatchTST (Nie et al., 2023) / MTST (Zhang et al., 2024) | ✓ | ✓ | × | × |
| iTransformer (Liu et al., 2024) | × | × | × | × |
| Medformer (Wang et al., 2024a) | ✓ | ✓ | × | × |
| **BioFormer (Ours)** | ✓ | ✓ | ✓ | ✓ |

Table 7 further compares BioFormer with representative DG-based methods. Unlike generic DG approaches that mainly enforce domain invariance, BioFormer explicitly models the structured spectral nature of subject variability through band-wise frequency alignment.

*Table 7.* Comparison with representative DG methods for cross-subject BTS analysis.

| Method | Statistical Modeling | Alignment | Plug-and-Play | No Subject Required | Spectral Modeling |
|---|:---:|:---:|:---:|:---:|:---:|
| GRL / DANN (Ganin et al., 2016) | × | ✓ | ✓ | × | × |
| MMD (Long et al., 2015) / CORAL (Sun & Saenko, 2016) | ✓ | ✓ | ✓ | ✓ | × |
| MixStyle (Zhou et al., 2021b) / DSU (Li et al., 2022) | ✓ | × | ✓ | ✓ | × |
| SubjectNorm (Rayatdoost et al., 2021) | ✓ | × | ✓ | × | × |
| DMMR (Wang et al., 2024b) | ✓ | ✓ | × | ✓ | × |
| SEDA-EEG (Tan et al., 2025) / CADAN (Huang et al., 2025) | × | ✓ | × | × | × |
| **BioFormer (Ours)** | ✓ | ✓ | ✓ | ✓ | ✓ |

Overall, BioFormer differs from existing DG methods by explicitly modeling the structured spectral form of subject variability, rather than treating cross-subject differences as generic domain discrepancy alone.

## E. Theorem: Fourier-Subspace Alignment via Amplitude/Phase Modulation

**Theorem**. Let $x \in \mathbb{R}^T$ be a real-valued time-series and let $\mathcal{F}$ denote the unitary DFT. For frequency bin $k$, denote the complex Fourier coefficient $z_k = (\mathcal{F}x)_k = A_k e^{j\phi_k}$ with amplitude $A_k \geq 0$ and phase $\phi_k \in (-\pi, \pi]$. Consider an *amplitude-phase modulation* that applies a per-bin complex multiplier

$$z_k' = u_k z_k, \qquad u_k := r_k e^{j\Delta_k}, \tag{12}$$

where $r_k \geq 0$ (amplitude scaling) and $\Delta_k$ (phase shift) may depend on the input sample. Then, for each frequency bin $k$, the mapping in (12) is exactly a *rotation + scaling* in the 2D real subspace spanned by $\{\cos(2\pi kt/T), \sin(2\pi kt/T)\}$. Moreover, for any chosen template spectrum $z_k^\star$ (e.g., a class-wise canonical spectrum), there exists a modulation $u_k$ such that $z_k' = z_k^\star$ for all samples with $z_k \neq 0$, thereby eliminating subject-induced spectral drift on those bins.

**Proof**

**Sin/cos subspace view.** Since $x$ is real, each frequency component can be written in the real Fourier form $x(t) = \sum_k \left(a_k^{(c)} \cos(2\pi kt/T) + a_k^{(s)} \sin(2\pi kt/T)\right)$. The complex coefficient $z_k$ is in one-to-one correspondence with the 2D vector $\mathbf{a}_k := [a_k^{(c)}, a_k^{(s)}]^\top$, and $(A_k, \phi_k)$ are simply its polar coordinates.

**Amplitude/phase modulation equals scaling+rotation.** Multiplying $z_k$ by $u_k = r_k e^{j\Delta_k}$ rotates the angle by $\Delta_k$ and scales the magnitude by $r_k$. In the real 2D coordinates, this is

$$\mathbf{a}_k' = r_k \begin{bmatrix} \cos\Delta_k & -\sin\Delta_k \\ \sin\Delta_k & \cos\Delta_k \end{bmatrix} \mathbf{a}_k, \tag{13}$$

which is precisely rotation + scaling in the $\{\cos, \sin\}$ subspace at frequency $k$.

**Existence of an aligning modulation.** Fix any template $z_k^\star$. For a sample with $z_k \neq 0$, choose $u_k := z_k^\star/z_k = (|z_k^\star|/A_k) e^{j(\arg z_k^\star - \phi_k)}$. Then $z_k' = u_k z_k = z_k^\star$ holds by construction, i.e., all samples are perfectly aligned on that frequency bin (up to bins where $A_k \approx 0$, which can be handled by an $\varepsilon$-stabilized ratio).

Finally, since $\mathcal{F}$ is unitary, aligning spectra implies aligning the corresponding projections in time: $\|x' - \tilde{x}'\|_2 = \|\mathcal{F}x' - \mathcal{F}\tilde{x}'\|_2$. Thus, amplitude/phase modulation provides a direct and structured mechanism to reduce cross-subject drift by aligning coefficients in Fourier subspaces.

## F. Frequency-Band Discriminability (FBD)

To quantitatively evaluate cross-subject generalization from a frequency-domain perspective, we define *Frequency-Band Discriminability (FBD)*, a metric that measures the relative strength of task-related spectral differences against subject-induced variability.

**Spectral Representation and Band Partition.** Let $z(t) \in \mathbb{R}^D$ denote the final-layer temporal embedding. For each channel $z_d(t)$, we compute the power spectral density (PSD) via the Fourier transform:

$$\text{PSD}_d(f) = |\mathcal{F}\{z_d(t)\}|^2. \tag{14}$$

The frequency range $[0, f_s/2]$ is partitioned into non-overlapping **2 Hz frequency bins**:

$$\mathcal{B} = \{b_k = [2k, 2k+2)\}. \tag{15}$$

For a given trial, the band power in bin $b$ is defined as

$$E_b = \frac{1}{D} \sum_{d=1}^{D} \int_{f \in b} \text{PSD}_d(f)\, df. \tag{16}$$

Band powers are further averaged across trials to obtain $\bar{E}_b^{(y,s)}$ for subject $s$ under class $y$.

**Intra- and Inter-Class Spectral Discrepancy.** The **intra-class discrepancy** measures subject-induced spectral variability within the same class:

$$\text{Intra}(b) = \frac{1}{C} \sum_{y=1}^{C} \text{Std}_s \left( \bar{E}_b^{(y,s)} \right). \tag{17}$$

The **inter-class discrepancy** quantifies task-related spectral separation. Let

$$\mu_b^{(y)} = \frac{1}{S_y} \sum_s \bar{E}_b^{(y,s)}, \tag{18}$$

then

$$\text{Inter}(b) = \max_y \mu_b^{(y)} - \min_y \mu_b^{(y)}. \tag{19}$$

**Frequency-Band Discriminability.** We define FBD as the ratio between inter- and intra-class discrepancies:

$$\boxed{\text{FBD}(b) = \frac{\text{Inter}(b)}{\text{Intra}(b) + \varepsilon}} \tag{20}$$

where $\varepsilon$ is a small constant for numerical stability.

An FBD value greater than 1 indicates that task-related spectral differences dominate subject-specific variability in band $b$, while values no greater than 1 suggest subject-driven variations.

**Band-Level Aggregation.** FBD is first computed at the **2 Hz bin level** for fine-grained analysis. To obtain a more stable summary, we report **band-level FBD** by averaging FBD values across adjacent bins within a broader frequency range. This aggregation mitigates the effect of isolated peaks caused by near-zero intra-class variance and provides a robust proxy for spectral structural alignment in cross-subject settings.

**Corollary F.1** (FBD Improvement via Intra-class Spectral Contraction). *Let* $\mathbf{a}_k(x) \in \mathbb{R}^2$ *denote the real Fourier-subspace coordinates at frequency bin $k$ (i.e., the coefficients on $\{\cos(2\pi kt/T), \sin(2\pi kt/T)\}$), and let $y \in \{1, \dots, C\}$ be the task label. Define the class-wise mean*

$$\boldsymbol{\mu}_{y,k} := \mathbb{E}[\mathbf{a}_k(x) \mid y],$$

*the intra-class discrepancy*

$$\text{Intra}_k := \mathbb{E}\left[\|\mathbf{a}_k(x) - \boldsymbol{\mu}_{y,k}\|_2^2\right],$$

*and the inter-class discrepancy (averaged over class pairs)*

$$\text{Inter}_k := \mathbb{E}_{y \neq y'}\left[\|\boldsymbol{\mu}_{y,k} - \boldsymbol{\mu}_{y',k}\|_2^2\right].$$

*Let* $\text{FBD}_k := \text{Inter}_k/\text{Intra}_k$.

*Consider a Fourier modulation $\mathbf{a}'_k(x) = r_k(x)\mathbf{R}(\Delta_k(x))\mathbf{a}_k(x)$ (as in Appendix E), and assume it satisfies:*

$$\text{Intra}'_k \leq \alpha_k \text{Intra}_k, \quad 0 < \alpha_k < 1, \qquad \text{Inter}'_k \geq \beta_k \text{Inter}_k, \quad \beta_k \geq 1.$$

*Then the FBD improves multiplicatively:*

$$\text{FBD}'_k \geq \frac{\beta_k}{\alpha_k} \text{FBD}_k.$$

*In particular, if inter-class structure is preserved ($\beta_k = 1$) while intra-class drift contracts, then $\text{FBD}'_k$ strictly increases.*

*Proof (sketch).* By definition,

$$\text{FBD}'_k = \frac{\text{Inter}'_k}{\text{Intra}'_k}.$$

Using the assumptions $\text{Inter}'_k \geq \beta_k \text{Inter}_k$ and $\text{Intra}'_k \leq \alpha_k \text{Intra}_k$, we obtain

$$\text{FBD}'_k \geq \frac{\beta_k \text{Inter}_k}{\alpha_k \text{Intra}_k} = \frac{\beta_k}{\alpha_k} \text{FBD}_k,$$

which proves the claim. $\qquad\square$

# G. Complete results

## G.1. Main Results

In the *Table 8*, we report the complete cross-subject evaluation results of all compared methods. For each dataset, we provide performance on all metrics (Accuracy, Precision, Recall, F1, AUROC, and AUPRC), together with the mean and standard deviation over multiple random seeds. These full tables complement the main paper by exposing both average performance and variability across runs, enabling a more comprehensive assessment of robustness under cross-subject evaluation.

## G.2. Visual Analysis of FBAM Alignment Effectiveness

To qualitatively examine the alignment behavior of FBAM, we visualize intermediate representations on the APAVA dataset in the time domain, as shown in Figure 10. The figure compares temporal embeddings before and after a single FBAM layer for samples sharing the same label but originating from different subjects. Before FBAM, time-domain representations exhibit substantial inter-subject variability, manifested as high-variance regions at specific temporal locations. Such variability reflects subject-dependent signal characteristics rather than label-relevant patterns. After applying FBAM, inter-subject variance is markedly reduced across the temporal axis for both label categories. Notably, this reduction is accompanied by a systematic reshaping of the temporal waveforms, suggesting that FBAM actively re-organizes subject-dependent variations rather than merely suppressing them, while still retaining label-relevant temporal patterns. These observations provide qualitative evidence that frequency-band alignment in FBAM translates into more consistent and subject-invariant representations in the time domain, supporting its effectiveness for cross-subject generalization.

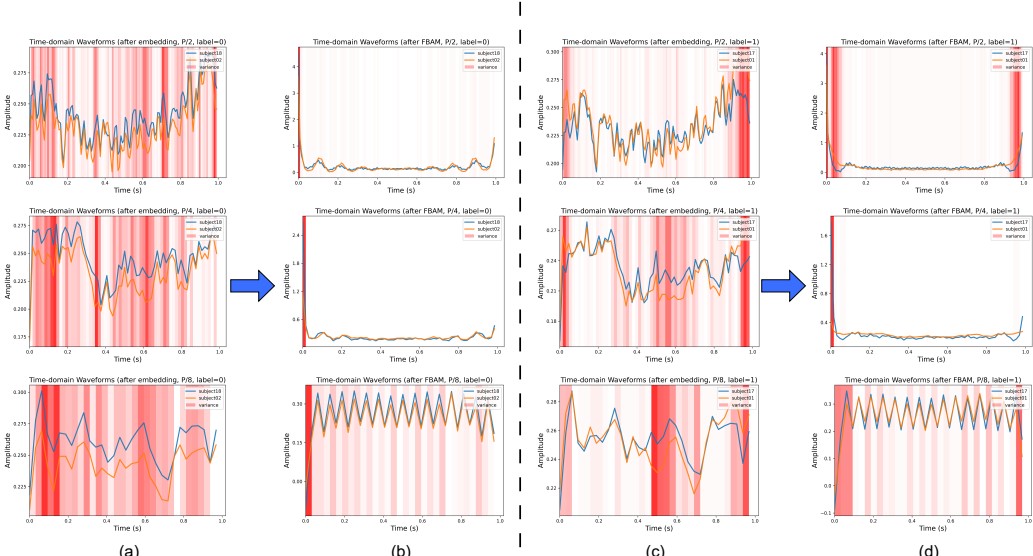

*Figure 10.* **Time-domain analysis of FBAM on the APAVA dataset.** Blue arrows indicate the temporal representations before and after a single FBAM layer. Panels (a) and (b) correspond to samples with label 0, while (c) and (d) correspond to label 1. Colored curves show time-domain waveforms from different subjects, and the red background encodes inter-subject variance, with darker regions indicating larger variance. Compared to the embeddings before FBAM, the representations after FBAM exhibit substantially reduced inter-subject variance in the time domain, suggesting that FBAM effectively suppresses subject-specific temporal discrepancies.

## G.3. The Analysis of FBAM

*Table 9* reports the ablation results on different configurations of the Frequency–Band Attention Module (FBAM) across four datasets. We investigate how each design choice affects cross-subject generalization in both EEG and ECG modalities.

**Custom Band Division (Variant 1).** We evaluate a variant that replaces uniform frequency band partitioning with dataset-specific band divisions. Figure 11 visualizes the frequency spectra of multi-scale tokens from APAVA, ADFTD, PTB, and PTB-XL at three temporal resolutions ($P/2$, $P/4$, $P/8$), where DC components are removed for clarity. Across datasets, low-frequency components dominate signal energy, while mid- and high-frequency regions exhibit dataset- and label-dependent variations.

Motivated by these observations, we manually define dataset-specific frequency bands to better reflect their average power

*Table 8.* Comparison with attention-based and shape-based time-series models under the *cross-subject* setup. Results are reported as mean ± std (%). Ranking uses only the mean (std is not considered). Best is highlighted with red background; second best with blue background. Avg. Rank is computed over six metrics (Acc/Prec/Rec/F1/AUROC/AUPRC), lower is better.

| Dataset | Model (Pub.) | Acc | Prec | Rec | F1 | AUROC | AUPRC | Avg. Rank↓ |
|---|---|---|---|---|---|---|---|---|
| **APAVA** (2 classes) | Transformer [NeurIPS'17] | 76.30±4.72 | 77.64±5.95 | 73.09±5.01 | 73.75±5.38 | 72.50±6.60 | 72.23±7.60 | 6.67 |
| | Reformer [ICLR'20] | 78.70±2.00 | 82.50±3.95 | 75.00±1.61 | 75.93±1.82 | 73.94±1.40 | 76.04±1.14 | 4.67 |
| | Informer [AAAI'21] | 73.11±4.40 | 75.17±6.06 | 69.17±4.56 | 69.47±5.06 | 70.46±4.91 | 70.75±5.27 | 9.83 |
| | Autoformer [NeurIPS'21] | 68.64±1.82 | 68.48±2.10 | 68.77±2.27 | 68.06±1.94 | 75.94±3.61 | 74.38±4.05 | 10.00 |
| | FEDformer [ICML'22] | 74.94±2.15 | 74.59±1.50 | 73.56±3.55 | 73.51±3.39 | 83.72±1.97 | 82.94±2.37 | 6.00 |
| | Nonformer [NeurIPS'22] | 71.89±3.81 | 71.80±4.58 | 69.44±3.56 | 69.74±3.84 | 70.55±2.96 | 70.78±4.08 | 9.83 |
| | PatchTST [ICLR'23] | 67.03±1.65 | 78.76±1.28 | 59.91±2.02 | 55.97±3.10 | 65.65±0.28 | 67.99±0.76 | 12.00 |
| | Crossformer [ICLR'23] | 73.77±1.95 | 79.29±4.36 | 68.86±1.70 | 68.93±1.85 | 72.39±3.33 | 72.05±3.65 | 8.67 |
| | iTransformer [ICLR'24] | 74.55±1.66 | 74.77±2.10 | 71.76±1.72 | 72.30±1.79 | 85.59±1.55 | 84.39±1.57 | 5.83 |
| | MTST [PMLR, 2024] | 71.14±1.59 | 79.30±2.97 | 65.27±2.28 | 64.01±3.16 | 68.87±2.34 | 71.06±1.60 | 10.33 |
| | Medformer [NeurIPS'24] | 78.74±0.64 | 81.11±0.84 | 75.40±0.66 | 76.31±0.71 | 83.20±0.91 | 83.66±0.92 | 3.50 |
| | SoftShape [ICML'25] | 81.22±0.62 | 85.65±0.22 | 77.55±0.87 | 78.71±0.91 | 82.24±2.43 | 83.93±2.19 | 2.67 |
| | BioFormer (Ours) | 82.31±1.55 | 87.87±0.67 | 78.50±1.94 | 79.77±2.09 | 88.52±2.25 | 88.16±2.40 | 1.00 |
| **ADFTD** (3 classes) | Transformer [NeurIPS'17] | 50.47±2.14 | 49.13±1.83 | 48.01±1.53 | 48.09±1.59 | 67.93±1.59 | 48.93±2.02 | 5.17 |
| | Reformer [ICLR'20] | 50.78±1.17 | 49.64±1.49 | 49.89±1.67 | 47.94±0.69 | 69.17±1.58 | 51.73±1.94 | 3.83 |
| | Informer [AAAI'21] | 48.45±1.96 | 46.54±1.68 | 46.06±1.84 | 45.74±1.38 | 65.87±1.27 | 47.60±1.30 | 8.33 |
| | Autoformer [NeurIPS'21] | 45.25±1.48 | 43.67±1.94 | 42.96±2.03 | 42.59±1.85 | 61.02±1.82 | 43.10±2.30 | 12.00 |
| | FEDformer [ICML'22] | 46.30±0.59 | 46.05±0.76 | 44.22±1.38 | 43.91±1.37 | 62.62±1.75 | 46.11±1.44 | 10.17 |
| | Nonformer [NeurIPS'22] | 49.95±1.05 | 47.71±0.97 | 47.46±1.50 | 46.96±1.35 | 66.23±1.37 | 47.33±1.78 | 7.17 |
| | PatchTST [ICLR'23] | 44.37±0.95 | 42.40±1.13 | 42.06±1.48 | 41.97±1.37 | 62.49±1.79 | 42.49±1.79 | 13.00 |
| | Crossformer [ICLR'23] | 50.45±2.31 | 45.57±1.63 | 45.88±1.82 | 45.50±1.70 | 66.45±2.03 | 48.33±2.05 | 8.17 |
| | iTransformer [ICLR'24] | 52.60±1.59 | 46.79±1.27 | 47.28±1.19 | 46.79±1.13 | 67.26±1.16 | 49.53±1.21 | 6.00 |
| | MTST [PMLR, 2024] | 45.60±2.03 | 44.70±1.33 | 45.05±1.30 | 44.31±1.74 | 62.50±0.81 | 45.16±0.85 | 10.67 |
| | Medformer [NeurIPS'24] | 53.27±1.54 | 51.02±1.57 | 50.71±1.55 | 50.65±1.51 | 70.93±1.19 | 51.21±1.32 | 2.33 |
| | SoftShape [ICML'25] | 54.60±0.82 | 50.33±1.24 | 49.17±0.89 | 48.91±1.13 | 70.08±0.98 | 50.90±1.12 | 3.17 |
| | BioFormer (Ours) | 56.73±2.51 | 54.66±2.59 | 54.60±3.02 | 53.82±3.03 | 74.74±2.08 | 58.07±2.72 | 1.00 |
| **PTB** (2 classes) | Transformer [NeurIPS'17] | 77.37±1.02 | 81.84±0.66 | 67.14±1.80 | 68.47±2.19 | 90.08±1.76 | 87.22±1.68 | 8.33 |
| | Reformer [ICLR'20] | 77.96±2.13 | 81.72±1.61 | 68.20±3.35 | 69.65±3.88 | 91.13±0.74 | 88.42±1.30 | 7.50 |
| | Informer [AAAI'21] | 78.69±1.68 | 82.87±1.02 | 69.19±2.90 | 70.84±3.47 | 92.09±0.53 | 90.02±0.60 | 5.67 |
| | Autoformer [NeurIPS'21] | 73.35±2.10 | 72.11±2.89 | 63.24±3.17 | 63.69±3.84 | 78.54±3.48 | 74.25±3.53 | 13.00 |
| | FEDformer [ICML'22] | 76.05±2.54 | 77.58±3.61 | 66.10±3.55 | 67.14±4.37 | 85.93±4.31 | 82.59±5.42 | 11.33 |
| | Nonformer [NeurIPS'22] | 78.66±0.49 | 82.77±0.86 | 69.12±0.87 | 70.90±1.00 | 89.37±2.51 | 86.67±2.38 | 7.33 |
| | PatchTST [ICLR'23] | 74.74±1.62 | 76.94±1.51 | 63.89±2.71 | 64.36±3.38 | 88.79±1.91 | 83.39±0.96 | 11.33 |
| | Crossformer [ICLR'23] | 80.17±3.79 | 85.04±1.83 | 71.25±6.29 | 72.75±7.19 | 88.55±3.45 | 87.31±3.25 | 6.17 |
| | iTransformer [ICLR'24] | 83.89±0.71 | 88.25±1.18 | 76.39±1.01 | 79.06±1.06 | 91.18±1.11 | 90.93±0.98 | 3.00 |
| | MTST [PMLR, 2024] | 76.59±1.90 | 79.88±1.90 | 66.31±2.95 | 67.38±3.71 | 86.86±2.75 | 83.75±2.84 | 10.17 |
| | Medformer [NeurIPS'24] | 83.50±2.01 | 85.19±0.94 | 77.11±3.39 | 79.18±3.31 | 92.81±1.48 | 90.32±1.54 | 3.17 |
| | SoftShape [ICML'25] | 83.72±0.83 | 86.70±0.67 | 75.74±1.28 | 78.42±1.30 | 94.38±0.76 | 93.00±0.88 | 3.00 |
| | BioFormer (Ours) | 87.73±1.73 | 88.71±1.28 | 82.63±2.80 | 84.71±2.51 | 94.71±0.94 | 93.53±0.96 | 1.00 |
| **PTB-XL** (5 classes) | Transformer [NeurIPS'17] | 70.59±0.44 | 61.57±0.65 | 57.62±0.35 | 59.05±0.25 | 88.21±0.16 | 63.36±0.29 | 9.67 |
| | Reformer [ICLR'20] | 71.72±0.43 | 63.12±1.02 | 59.20±0.75 | 60.69±0.18 | 88.80±0.24 | 64.72±0.47 | 7.17 |
| | Informer [AAAI'21] | 71.43±0.32 | 62.64±0.60 | 59.12±0.47 | 60.44±0.43 | 88.65±0.19 | 64.76±0.17 | 7.83 |
| | Autoformer [NeurIPS'21] | 61.68±2.72 | 51.60±1.64 | 49.10±1.52 | 48.85±2.27 | 82.04±1.44 | 51.93±1.71 | 12.50 |
| | FEDformer [ICML'22] | 57.20±9.47 | 52.38±6.09 | 49.04±7.26 | 47.89±8.44 | 82.13±4.17 | 52.31±7.03 | 12.50 |
| | Nonformer [NeurIPS'22] | 70.56±0.55 | 61.57±0.36 | 57.75±0.72 | 59.10±0.66 | 88.32±0.36 | 63.40±0.79 | 9.17 |
| | PatchTST [ICLR'23] | 73.23±0.25 | 65.70±0.64 | 60.82±0.76 | 62.61±0.34 | 89.74±0.19 | 67.32±0.22 | 3.00 |
| | Crossformer [ICLR'23] | 73.30±0.14 | 65.06±0.35 | 61.23±0.33 | 62.59±0.14 | 90.02±0.06 | 67.43±0.22 | 3.00 |
| | iTransformer [ICLR'24] | 69.28±0.22 | 59.59±0.45 | 54.62±0.18 | 56.20±0.19 | 86.71±0.10 | 60.27±0.21 | 11.00 |
| | MTST [PMLR, 2024] | 72.14±0.27 | 63.84±0.72 | 60.01±0.81 | 61.43±0.38 | 88.97±0.33 | 65.83±0.51 | 6.00 |
| | Medformer [NeurIPS'24] | 72.87±0.23 | 64.14±0.42 | 60.60±0.46 | 62.02±0.37 | 89.66±0.13 | 66.39±0.22 | 5.00 |
| | SoftShape [ICML'25] | 73.94±0.20 | 65.67±0.20 | 61.97±0.52 | 62.97±0.52 | 90.59±0.09 | 68.46±0.24 | 1.17 |
| | BioFormer (Ours) | 73.37±0.39 | 65.62±0.52 | 60.81±0.36 | 62.56±0.30 | 90.37±0.12 | 67.61±0.35 | 2.83 |
| **BCI-2a** (4 classes) | Transformer [NeurIPS'17] | 41.68±2.28 | 42.86±1.69 | 41.68±2.28 | 41.17±2.50 | 68.88±1.34 | 43.37±1.85 | 8.17 |
| | Reformer [ICLR'20] | 42.69±0.91 | 43.05±1.18 | 42.69±0.91 | 42.52±0.94 | 69.64±1.20 | 44.15±1.56 | 2.67 |
| | Informer [AAAI'21] | 42.52±1.10 | 43.24±1.21 | 42.52±1.10 | 42.24±0.99 | 69.20±1.27 | 43.72±1.69 | 4.50 |
| | Autoformer [NeurIPS'21] | 27.43±0.72 | 27.33±0.82 | 27.43±0.72 | 27.05±0.79 | 53.04±0.43 | 27.16±0.31 | 13.00 |
| | FEDformer [ICML'22] | 29.06±1.00 | 29.17±1.12 | 29.06±1.00 | 28.72±1.04 | 54.58±0.25 | 28.75±0.19 | 12.00 |
| | Nonformer [NeurIPS'22] | 42.67±0.63 | 43.60±0.95 | 42.67±0.63 | 42.51±0.70 | 69.31±0.49 | 43.98±0.67 | 3.17 |
| | PatchTST [ICLR'23] | 40.64±1.11 | 40.78±1.29 | 40.64±1.11 | 40.13±1.30 | 67.63±0.76 | 41.05±0.88 | 8.83 |
| | Crossformer [ICLR'23] | 42.48±2.90 | 43.95±3.47 | 42.48±2.90 | 41.61±3.05 | 69.22±2.75 | 43.44±3.16 | 4.83 |
| | iTransformer [ICLR'24] | 33.58±1.44 | 33.88±1.35 | 33.58±1.44 | 33.23±1.47 | 59.57±0.75 | 32.87±0.87 | 11.00 |
| | MTST [PMLR, 2024] | 39.31±0.99 | 39.72±0.77 | 39.31±0.99 | 39.06±1.11 | 66.61±0.55 | 39.79±0.46 | 9.83 |
| | Medformer [NeurIPS'24] | 42.40±0.69 | 42.90±0.72 | 42.40±0.69 | 42.29±0.72 | 69.09±0.23 | 43.51±0.31 | 4.83 |
| | SoftShape [ICML'25] | 40.87±7.22 | 41.68±7.35 | 40.87±7.22 | 38.69±8.67 | 68.57±6.07 | 44.23±8.13 | 8.17 |
| | BioFormer (Ours) | 46.68±1.56 | 47.24±1.64 | 46.68±1.56 | 46.35±1.59 | 72.49±1.12 | 47.63±1.53 | 1.00 |
| **BCI-2b** (2 classes) | Transformer [NeurIPS'17] | 70.86±1.22 | 70.96±1.28 | 70.86±1.22 | 70.83±1.20 | 77.82±1.63 | 77.23±1.74 | 6.67 |
| | Reformer [ICLR'20] | 70.90±0.46 | 71.13±0.37 | 70.90±0.46 | 70.82±0.50 | 78.34±0.80 | 77.73±0.98 | 5.67 |
| | Informer [AAAI'21] | 70.57±0.68 | 70.64±0.66 | 70.57±0.68 | 70.54±0.69 | 77.86±0.70 | 77.07±0.73 | 8.33 |
| | Autoformer [NeurIPS'21] | 58.00±0.47 | 58.04±0.51 | 58.00±0.47 | 57.95±0.44 | 61.06±0.62 | 59.46±0.73 | 13.00 |
| | FEDformer [ICML'22] | 60.99±1.64 | 61.18±1.66 | 60.99±1.64 | 60.82±1.68 | 66.19±2.48 | 65.37±2.57 | 12.00 |
| | Nonformer [NeurIPS'22] | 70.71±0.82 | 71.02±0.79 | 70.71±0.82 | 70.60±0.86 | 78.79±0.92 | 78.37±0.92 | 6.00 |
| | PatchTST [ICLR'23] | 73.26±0.65 | 73.61±0.69 | 73.26±0.65 | 73.17±0.68 | 81.69±0.33 | 81.36±0.41 | 1.00 |
| | Crossformer [ICLR'23] | 70.58±1.21 | 71.16±1.57 | 70.58±1.21 | 70.39±1.14 | 77.54±1.03 | 76.88±0.76 | 8.00 |
| | iTransformer [ICLR'24] | 65.94±0.61 | 66.15±0.68 | 65.94±0.61 | 65.84±0.63 | 72.17±1.26 | 71.53±1.43 | 11.00 |
| | MTST [PMLR, 2024] | 71.53±0.56 | 71.63±0.61 | 71.53±0.56 | 71.50±0.55 | 79.48±0.36 | 79.27±0.45 | 3.00 |
| | Medformer [NeurIPS'24] | 69.40±0.35 | 69.66±0.33 | 69.40±0.35 | 69.30±0.36 | 76.03±0.45 | 75.30±0.40 | 10.00 |
| | SoftShape [ICML'25] | 71.28±1.02 | 71.61±0.95 | 71.28±1.02 | 71.16±1.08 | 78.48±1.66 | 77.88±1.86 | 4.33 |
| | BioFormer (Ours) | 72.83±1.12 | 72.92±1.08 | 72.83±1.12 | 72.81±1.13 | 80.38±1.04 | 79.98±1.22 | 2.00 |

distributions. The resulting band boundaries for each dataset and scale are summarized below:

- APAVA: $[0,5,12,20,32,48,64], [0,5,12,20,32], [0,4,8,12,16]$.
- ADFTD: $[0,5,9,12,32,48,64], [0,4,7,12,32], [0,3,8,12,16]$.
- PTB: $[0,5,10,15,20,48,75], [0,5,10,20,37], [0,3,6,10,19]$.
- PTB-XL: $[0,4,8,15,20,48,62], [0,3,5,10,31], [0,2,6,9,16]$.

Although these custom band divisions slightly improve adaptability, the overall gains remain marginal. This suggests that uniform band partitioning already provides a robust and effective prior for FBAM across diverse biomedical datasets.

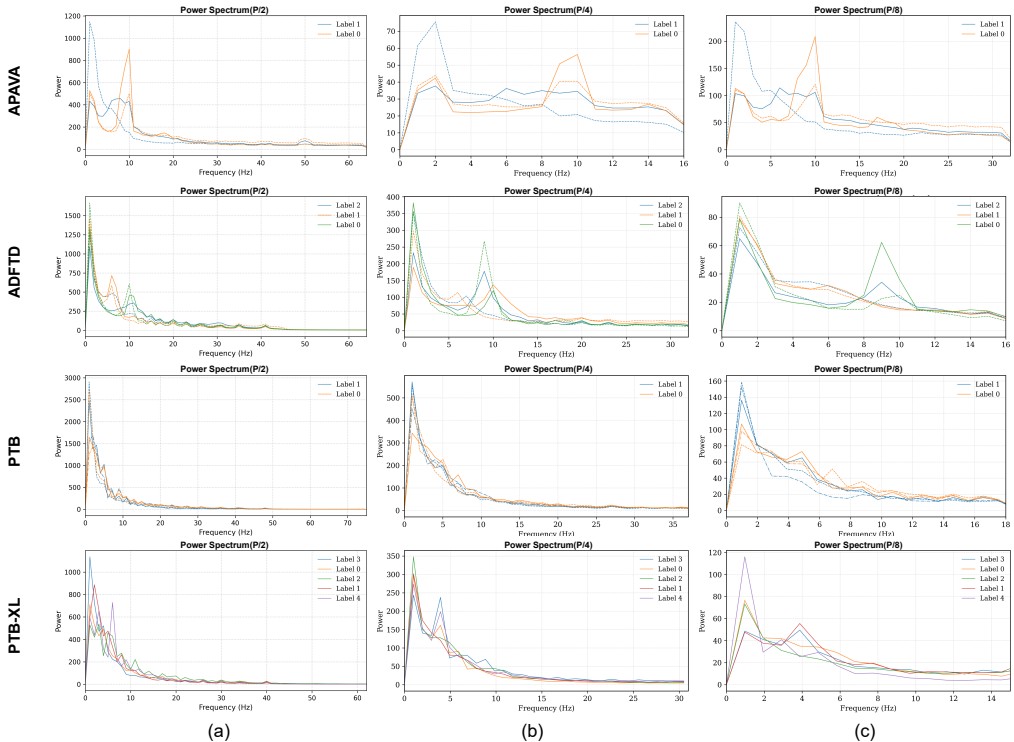

*Figure 11.* Frequency spectrum visualization across three pyramid scales:(a) $P/2$, (b) $P/4$, and (c) $P/8$.Different line styles represent objects with identical labels. It can be observed that low-frequency energy dominates across all datasets, while mid-to-high-frequency responses exhibit differences between modalities.

**DC Learning (Variant 2).** This variant introduces learnable weights for the DC component instead of keeping it fixed. Across datasets, this modification yields consistent performance gains, with particularly notable improvements on PTB. This suggests that low-frequency baseline dynamics, while often treated as nuisance, can encode discriminative information when adaptively modeled.

**Magnitude Residual (Variant 3).** This variant removes the residual formulation and directly applies magnitude-domain convolution. Performance degrades consistently across all datasets, indicating that residual correction is critical for stabilizing frequency responses and preventing over-distortion during spectral modulation.

**Static Band Modulation (Variant 4).** This variant replaces the cross-attention–driven parameter generation with static, learnable band-wise parameters. Specifically, the modulation coefficients for magnitude and phase, including the dynamic convolution kernel $w_k$, residual gain $g_k$, and phase offset $\beta_k$, are initialized as free parameters and optimized directly, without conditioning on band-level representations. As shown in *Table 9*, this modification leads to mixed performance changes across datasets, with noticeable degradation on ADFTD and limited gains on PTB-XL. These results indicate that fixed band-wise modulation lacks sufficient adaptivity to accommodate subject-dependent spectral variations, highlighting the importance of data-driven parameter generation conditioned on spectral context.

**Intra-Band Self-Attention (Variant 5).** This variant replaces the cross-attention mechanism between learnable band tokens and band statistics with a self-attention operation performed directly among band representations. In this setting,

band interactions are modeled symmetrically without an explicit token-based query structure. While this design preserves interactions across frequency bands, it consistently underperforms the cross-attention formulation in *Table 9*, especially on APAVA and ADFTD. The results suggest that cross-attention with dedicated band tokens provides a more effective inductive bias for extracting global modulation cues from band-level summaries than self-attention alone.

*Table 9.* Ablation study on different configurations of the Frequency–Band Attention Module (FBAM). Columns correspond to four datasets, each reporting Accuracy and F1-score (%, mean ± std). The best results for each dataset are highlighted in **bold**.

| Model Variant | APAVA | | PTB | | ADFTD | | PTB-XL | |
|---|---|---|---|---|---|---|---|---|
| | Acc. | F1 | Acc. | F1 | Acc. | F1 | Acc. | F1 |
| Variant 1 | 80.07±2.19 | 76.83±3.20 | 84.81±1.58 | 80.58±2.59 | 55.59±1.02 | 52.52±0.56 | 73.15±0.49 | 62.50±0.69 |
| Variant 2 | 80.89±2.05 | 78.08±2.64 | 86.64±1.23 | 83.40±1.79 | **56.83±2.97** | 53.59±3.31 | 73.00±0.48 | 62.14±0.48 |
| Variant 3 | 71.05±5.88 | 68.74±4.40 | 84.37±1.20 | 80.09±1.90 | 55.93±1.14 | 53.40±1.03 | 72.72±0.73 | 61.42±1.01 |
| Variant 4 | 81.65±0.89 | 78.92±1.21 | 85.36±0.87 | 81.28±1.35 | 51.97±1.42 | 48.28±1.17 | 72.80±0.26 | **62.91±0.33** |
| Variant 5 | 79.57±1.26 | 76.11±1.79 | 86.82±1.30 | 83.49±1.98 | 54.77±3.34 | 51.42±3.40 | 72.91±0.38 | 61.88±0.34 |
| **BioFormer (Ours)** | **82.31±1.55** | **79.77±2.09** | **87.73±1.73** | **84.71±2.51** | 56.73±2.51 | **53.82±3.03** | **73.37±0.39** | 62.56±0.30 |

Finally, **BioFormer** achieves the best overall performance. Results from Variants 1–5 indicate that simple uniform frequency-band partitioning is sufficient for FBAM, and that explicit learning of the DC component is unnecessary, while residual magnitude correction, input-conditioned band modulation, and cross-band attention each play a critical role in enabling effective frequency-domain alignment.

## G.4. Ablation on Band-Level Statistics for the Frequency Descriptor.

FBAM summarizes each frequency band using a compact descriptor $\mathbf{b}_k$ to characterize band-wise distributional states and guide adaptive modulation. *Table 10* evaluates different choices of band-level statistics under subject-independent evaluation. Using only low-order statistics (*LogMom*: log-mean, log-variance, and peak magnitude) already yields strong and stable performance across datasets, indicating that major cross-subject drift can be effectively captured by coarse distributional cues. Augmenting *LogMom* with peak location or total band energy provides additional structural information but often leads to inferior or less stable results, suggesting sensitivity to frequency discretization (*PeakLoc*) or excessive coarsening (*BandEnergy*).

In contrast, **BioFormer**, which combines five complementary statistics—log-mean, log-variance, peak magnitude, peak location, and band energy—and integrates them through cross-band attention, achieves the best overall performance on APAVA and PTB while remaining competitive on ADFTD. These results support our design choice of leveraging a richer yet structured statistical descriptor together with learnable inter-band modeling, rather than relying on a single handcrafted summary.

*Table 10.* Performance comparison under subject-independent evaluation. Different variants correspond to different choices of band-level statistics used to construct the frequency descriptor $\mathbf{b}_k$. *LogMom* uses $\{\log \mu_k, \log \sigma_k, \text{peak}_k\}$; *LogMom+PeakLoc* further incorporates the peak frequency location; *LogMom+BandEnergy* replaces the peak location with the total band energy. Accuracy and F1-score (%) are reported as mean ± standard deviation. Best results for each dataset are highlighted in **bold**.

| Band Descriptor | APAVA | | PTB | | ADFTD | |
|---|---|---|---|---|---|---|
| | Acc. | F1 | Acc. | F1 | Acc. | F1 |
| LogMom | 80.87±0.88 | 77.80±1.16 | 85.42±2.70 | 81.16±4.39 | **57.16±2.53** | **53.97±2.11** |
| LogMom+PeakLoc | 79.11±2.10 | 75.39±2.96 | 86.31±2.83 | 82.70±4.00 | 55.87±1.36 | 51.63±1.92 |
| LogMom+BandEnergy | 77.78±4.83 | 73.07±7.68 | 85.64±1.39 | 81.70±2.13 | 55.53±2.37 | 51.76±3.02 |
| BioFormer (Ours) | **82.31±1.55** | **79.77±2.09** | **87.73±1.73** | **84.71±2.51** | 56.73±2.51 | 53.82±3.03 |

## G.5. Comparison with BTS-specific DG Methods

To further contextualize BioFormer with respect to BTS-specific domain generalization methods, we additionally compare against DMMR (Wang et al., 2024b), a representative framework designed for cross-subject physiological signal analysis.

Table 11 reports the results under the same cross-subject evaluation protocol. BioFormer consistently outperforms DMMR across all evaluated datasets. In particular, the performance gap is substantial on APAVA and ADFTD, indicating that explicit spectral-structure modeling provides stronger robustness under limited-data settings.

We note that DMMR relies on a self-supervised reconstruction-based pretraining strategy, and its performance is therefore more sensitive to the availability of sufficiently large-scale physiological data. In contrast, BioFormer is trained end-to-end

using only the downstream classification objective without additional pretraining.

*Table 11.* Comparison with the BTS-specific DG method DMMR.

| Dataset | DMMR Acc. | DMMR F1 | BioFormer Acc. | BioFormer F1 |
|---------|-----------|---------|----------------|--------------|
| APAVA   | 64.74     | 63.45   | **82.31**      | **79.77**    |
| PTB     | 77.06     | 68.31   | **87.73**      | **84.71**    |
| ADFTD   | 52.21     | 40.83   | **56.73**      | **53.82**    |
| PTB-XL  | 69.53     | 53.14   | **73.37**      | **62.56**    |

## G.6. Additional Analysis on Subject-specific Variability

To complement the subject separability analysis in Sec. 5.5, we further evaluate silhouette statistics on the learned representations extracted from unseen test subjects.

We report: (i) **Label Silhouette**, which measures task-class separability, and (ii) **Subject Silhouette**, which measures subject-wise clustering tendency. Higher Label Silhouette indicates stronger semantic structure, while lower Subject Silhouette indicates weaker subject dependency.

*Table 12.* **Silhouette analysis of learned representations.** Higher Label Silhouette indicates better task separability, while lower Subject Silhouette indicates weaker subject-specific clustering.

| Dataset | Metric | Trans.+GRL | SoftShape | BioFormer |
|---------|--------|------------|-----------|-----------|
| APAVA   | Label ↑   | 0.1137  | 0.1699   | **0.3035**  |
|         | Subject ↓ | 0.3915  | 0.4002   | **0.2904**  |
| PTB     | Label ↑   | 0.3820  | 0.4727   | **0.4783**  |
|         | Subject ↓ | -0.2442 | **-0.2468** | -0.2452  |

As shown in Table 12, BioFormer consistently achieves stronger task-related clustering while maintaining weak subject-wise grouping. On APAVA, BioFormer obtains both the highest Label Silhouette and the lowest Subject Silhouette, indicating improved task-centric organization with reduced subject dependency. On PTB, all methods exhibit weak subject clustering, while BioFormer still preserves the strongest task separability without explicit subject supervision.

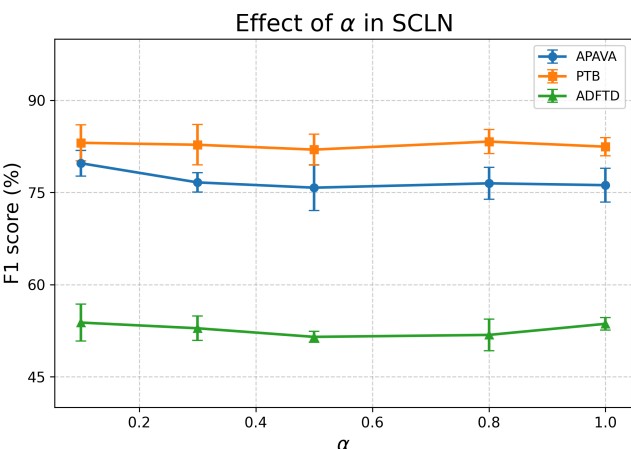

*Figure 12.* Effect of the weighting factor $\alpha$ in SCLN on different datasets. F1 scores are reported as the mean $\pm$ standard deviation over five independent runs.

## G.7. Sensitivity Analysis of $\alpha$ in SCLN

In Sample Conditional Layer Normalization (SCLN), the weighting factor $\alpha \in [0, 1]$ controls the interpolation between the original representation and the sample-adaptively normalized one:

$$\mathbf{h}_{\text{out}} = (1 - \alpha)\mathbf{LN}(h) + \alpha\big(\boldsymbol{\gamma} \odot \text{LN}(\mathbf{h}) + \boldsymbol{\beta}\big). \tag{21}$$

As shown in Figure 12, the performance exhibits a non-monotonic dependency on $\alpha$. Specifically, stronger performance is

consistently observed when $\alpha$ lies in the lower range (0–0.3) or the higher range (0.8–1.0), while intermediate values tend to degrade performance across datasets. This behavior suggests two effective regimes of SCLN. When $\alpha$ is small, SCLN acts as a mild regularizer that preserves the original encoder representation while providing limited correction for sample-level drift. When $\alpha$ is large, normalization dominates and aggressively suppresses residual distributional variations introduced by FBAM. In contrast, intermediate values of $\alpha$ may be insufficient to fully remove drift while simultaneously perturbing discriminative structure, leading to suboptimal trade-offs. Overall, this analysis indicates that SCLN is robust across a wide range of settings and that its benefit arises from either minimally invasive calibration or strong normalization, rather than partial modulation.

### G.8. Computational Complexity Comparison

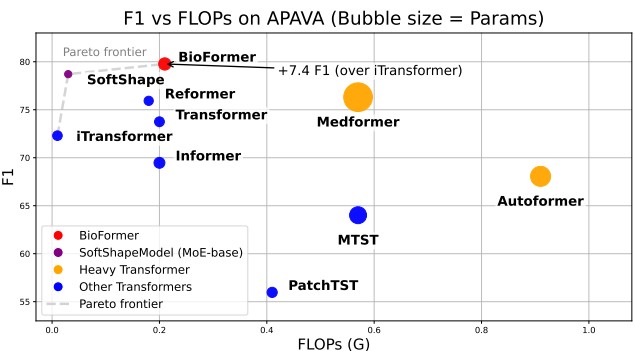

*Figure 13.* **The classification performance and computational complexity on the APAVA dataset.** All FLOPs are measured with an input tensor of shape $(1, 256, 16)$.

We report the parameter count and FLOPs of all methods on the APAVA dataset to analyze the performance–efficiency trade-off, with the complete results presented in Table 13. As shown in Figure 13, BioFormer achieves the best F1-score while maintaining a low parameter count and moderate FLOPs, demonstrating strong efficiency. Compared with the standard Transformer backbone, FBAM brings an absolute F1 improvement of about 6%, without increasing the model scale. Heavier Transformer-based models rely on substantially higher computational cost, but do not yield better performance. SoftShape Model, a CNN-style mixture-of-experts architecture, is parameter-efficient but still underperforms BioFormer, highlighting the benefit of frequency-band alignment within Transformer models.

*Table 13.* Comparison of performance and model complexity on the APAVA dataset under the cross-subject setting. F1-score (%) is reported. Model complexity is measured by the number of parameters (M) and FLOPs per sample (G). All FLOPs are measured with an input tensor of shape $(1, 256, 16)$.

| Model | F1 (%) | Params (M) | FLOPs (G) |
| --- | --- | --- | --- |
| Transformer | 73.75 | 0.87 | 0.20 |
| Reformer | 75.93 | 0.77 | 0.18 |
| Informer | 69.47 | 1.07 | 0.20 |
| Autoformer | 68.06 | 3.61 | 0.91 |
| FEDformer | 73.51 | 1.46 | 0.20 |
| Nonstationary | 69.74 | 0.94 | 0.20 |
| PatchTST | 55.97 | 0.93 | 0.41 |
| Crossformer | 68.93 | 5.23 | 0.21 |
| iTransformer | 72.30 | 0.83 | 0.01 |
| MTST | 64.01 | 2.59 | 0.57 |
| Medformer | 76.31 | 7.41 | 0.57 |
| SoftShape | 78.71 | 0.47 | 0.03 |
| **BioFormer** | **79.77** | **1.36** | **0.21** |

### G.9. Subject-dependent Evaluation

In addition to the cross-subject setting, we evaluate BioFormer under the **subject-dependent** protocol, where training and testing samples may originate from the same subjects. *Table 14* presents a comprehensive comparison with attention-based and shape-based time-series models across four datasets. As expected, most methods achieve substantially higher performance in this setting due to the reduced distribution shift, and strong time-domain or shape-based baselines such as SoftShape and Nonformer perform competitively, particularly on APAVA and PTB.

Despite being designed for cross-subject robustness, BioFormer remains consistently competitive under subject-dependent evaluation, achieving top or near-top average ranks on APAVA, ADFTD, and PTB, and maintaining strong performance on the large-scale PTB-XL dataset. These results indicate that BioFormer does not trade off performance in the easier subject-dependent regime, while offering clear advantages under subject-independent evaluation, demonstrating its robustness across different evaluation protocols.

*Table 14.* Comparison with attention-based and shape-based time-series models under the *subject-dependent* setup. Results are reported as mean ± std (%). Ranking uses only the mean (std is not considered). Best is highlighted with red background; second best with blue background. Avg. Rank is computed over six metrics (Acc/Prec/Rec/F1/AUROC/AUPRC), lower is better.

| Dataset | Model (Pub.) | Acc | Prec | Rec | F1 | AUROC | AUPRC | Avg. Rank↓ |
|---|---|---|---|---|---|---|---|---|
| **APAVA** (2 classes) | Transformer [NeurIPS'17] | 98.51±0.19 | 98.32±0.22 | 98.58±0.20 | 98.45±0.20 | 99.92±0.03 | 99.91±0.03 | 5.83 |
| | Reformer [ICLR'20] | 98.64±0.32 | 98.69±0.29 | 98.48±0.40 | 98.58±0.33 | 99.91±0.03 | 99.91±0.03 | 5.17 |
| | Informer [AAAI'21] | 98.86±0.17 | 98.81±0.18 | 98.81±0.20 | 98.81±0.18 | 99.91±0.03 | 99.90±0.03 | 4.50 |
| | Autoformer [NeurIPS'21] | 98.53±0.56 | 98.43±0.67 | 98.50±0.51 | 98.46±0.58 | 99.73±0.18 | 99.71±0.20 | 6.67 |
| | FEDformer [ICML'22] | 92.85±0.60 | 93.18±0.50 | 91.88±0.77 | 92.42±0.66 | 98.28±0.13 | 98.25±0.12 | 12.50 |
| | Nonformer [NeurIPS'22] | 99.01±0.20 | 98.87±0.22 | 99.07±0.20 | 98.97±0.21 | 99.94±0.03 | 99.93±0.03 | 3.00 |
| | PatchTST [ICLR'23] | 97.55±0.26 | 97.61±0.28 | 97.28±0.29 | 97.44±0.27 | 99.67±0.04 | 99.63±0.05 | 10.50 |
| | Crossformer [ICLR'23] | 97.60±0.61 | 97.51±0.65 | 97.49±0.63 | 97.50±0.63 | 99.70±0.08 | 99.70±0.08 | 9.67 |
| | iTransformer [ICLR'24] | 93.20±1.06 | 92.76±1.08 | 93.18±1.18 | 92.93±1.11 | 98.11±0.42 | 97.94±0.52 | 12.50 |
| | MTST [PMLR, 2024] | 97.62±0.29 | 97.51±0.33 | 97.53±0.32 | 97.52±0.31 | 99.68±0.06 | 99.58±0.08 | 9.67 |
| | Medformer [NeurIPS'24] | 98.26±0.23 | 98.34±0.28 | 98.02±0.21 | 98.18±0.24 | 99.81±0.03 | 99.78±0.05 | 7.50 |
| | SoftShape [ICML'25] | 99.92±0.00 | 99.92±0.02 | 99.91±0.02 | 99.91±0.00 | 100.00±0.00 | 100.00±0.00 | 1.00 |
| | BioFormer (Ours) | 99.50±0.14 | 99.52±0.14 | 99.43±0.16 | 99.47±0.15 | 99.99±0.01 | 99.99±0.01 | 2.00 |
| **ADFTD** (3 classes) | Transformer [NeurIPS'17] | 97.14±0.41 | 97.02±0.44 | 97.02±0.40 | 97.02±0.42 | 99.76±0.05 | 99.43±0.11 | 1.00 |
| | Reformer [ICLR'20] | 89.75±2.90 | 89.85±2.78 | 89.86±2.55 | 89.46±3.01 | 98.30±0.75 | 96.93±1.29 | 4.67 |
| | Informer [AAAI'21] | 91.18±0.89 | 90.85±1.02 | 91.06±0.80 | 90.91±0.91 | 98.29±0.27 | 96.73±0.49 | 4.33 |
| | Autoformer [NeurIPS'21] | 84.98±3.12 | 84.85±3.16 | 84.18±3.32 | 84.40±3.27 | 95.12±1.66 | 91.33±2.78 | 8.33 |
| | FEDformer [ICML'22] | 84.18±0.46 | 83.64±0.62 | 83.56±0.27 | 83.58±0.40 | 95.36±0.20 | 91.60±0.36 | 8.67 |
| | Nonformer [NeurIPS'22] | 95.89±0.31 | 95.67±0.37 | 95.77±0.32 | 95.72±0.33 | 99.57±0.06 | 99.05±0.14 | 3.00 |
| | PatchTST [ICLR'23] | 66.35±0.45 | 65.36±0.46 | 65.15±0.40 | 65.11±0.35 | 83.24±0.41 | 72.36±0.70 | 11.00 |
| | Crossformer [ICLR'23] | 88.98±1.41 | 88.69±1.47 | 88.47±1.45 | 88.56±1.47 | 97.44±0.59 | 95.36±1.03 | 6.17 |
| | iTransformer [ICLR'24] | 64.75±0.61 | 62.69±0.51 | 62.53±0.36 | 62.49±0.42 | 81.49±0.15 | 69.02±0.26 | 13.00 |
| | MTST [PMLR, 2024] | 65.07±0.55 | 64.27±0.86 | 63.28±0.26 | 63.49±0.35 | 81.51±0.44 | 69.77±0.64 | 12.00 |
| | Medformer [NeurIPS'24] | 69.58±0.67 | 72.92±1.07 | 67.04±0.81 | 67.64±0.79 | 88.04±0.52 | 79.79±0.81 | 10.00 |
| | SoftShape [ICML'25] | 88.82±0.29 | 88.21±0.32 | 88.27±0.37 | 88.22±0.32 | 97.54±0.10 | 95.13±0.22 | 6.83 |
| | BioFormer (Ours) | 96.72±0.91 | 96.53±0.92 | 96.63±0.96 | 96.58±0.94 | 99.71±0.13 | 99.42±0.25 | 2.00 |
| **PTB** (2 classes) | Transformer [NeurIPS'17] | 99.90±0.02 | 99.85±0.05 | 99.79±0.03 | 99.82±0.03 | 99.88±0.00 | 99.89±0.01 | 4.17 |
| | Reformer [ICLR'20] | 99.87±0.01 | 99.85±0.05 | 99.69±0.02 | 99.77±0.02 | 99.90±0.01 | 99.89±0.01 | 5.67 |
| | Informer [AAAI'21] | 99.86±0.02 | 99.82±0.05 | 99.68±0.02 | 99.75±0.03 | 99.86±0.01 | 99.88±0.01 | 8.00 |
| | Autoformer [NeurIPS'21] | 99.77±0.15 | 99.59±0.45 | 99.59±0.12 | 99.59±0.28 | 99.94±0.02 | 99.92±0.01 | 9.33 |
| | FEDformer [ICML'22] | 99.81±0.01 | 99.77±0.05 | 99.55±0.08 | 99.66±0.02 | 99.93±0.04 | 99.91±0.03 | 9.00 |
| | Nonformer [NeurIPS'22] | 99.90±0.01 | 99.86±0.02 | 99.76±0.02 | 99.81±0.02 | 99.88±0.02 | 99.89±0.01 | 4.67 |
| | PatchTST [ICLR'23] | 99.85±0.02 | 99.80±0.06 | 99.65±0.02 | 99.72±0.03 | 99.94±0.02 | 99.92±0.02 | 6.33 |
| | Crossformer [ICLR'23] | 99.84±0.03 | 99.78±0.08 | 99.62±0.05 | 99.70±0.05 | 99.98±0.01 | 99.97±0.01 | 6.67 |
| | iTransformer [ICLR'24] | 99.80±0.02 | 99.75±0.03 | 99.52±0.04 | 99.64±0.03 | 99.99±0.01 | 99.97±0.01 | 8.33 |
| | MTST [PMLR, 2024] | 99.83±0.01 | 99.72±0.02 | 99.65±0.04 | 99.69±0.03 | 99.90±0.02 | 99.90±0.03 | 9.17 |
| | Medformer [NeurIPS'24] | 99.86±0.04 | 99.81±0.05 | 99.68±0.04 | 99.75±0.04 | 99.92±0.01 | 99.91±0.01 | 6.00 |
| | SoftShape [ICML'25] | 99.90±0.01 | 99.86±0.03 | 99.78±0.01 | 99.82±0.02 | 99.86±0.03 | 99.88±0.02 | 4.83 |
| | BioFormer (Ours) | 99.90±0.02 | 99.85±0.06 | 99.77±0.03 | 99.81±0.05 | 99.90±0.02 | 99.91±0.02 | 3.67 |
| **PTB-XL** (5 classes) | Transformer [NeurIPS'17] | 88.38±0.34 | 85.51±0.61 | 84.90±0.47 | 85.18±0.45 | 97.85±0.09 | 91.71±0.34 | 2.00 |
| | Reformer [ICLR'20] | 77.43±0.90 | 70.71±1.41 | 68.18±1.56 | 69.24±1.43 | 93.07±0.50 | 75.35±1.47 | 6.00 |
| | Informer [AAAI'21] | 75.12±0.37 | 67.42±0.59 | 65.47±0.97 | 66.26±0.72 | 91.55±0.25 | 71.51±0.75 | 8.33 |
| | Autoformer [NeurIPS'21] | 64.42±3.56 | 55.15±4.01 | 54.55±3.68 | 53.97±4.15 | 85.33±2.18 | 57.95±4.17 | 12.50 |
| | FEDformer [ICML'22] | 59.90±9.91 | 56.74±5.92 | 53.78±7.01 | 52.46±8.17 | 85.37±4.24 | 58.61±7.23 | 12.50 |
| | Nonformer [NeurIPS'22] | 89.10±0.55 | 86.77±0.73 | 85.30±0.93 | 86.00±0.78 | 98.02±0.15 | 92.31±0.60 | 1.00 |
| | PatchTST [ICLR'23] | 75.44±0.26 | 67.82±0.53 | 64.72±0.40 | 66.03±0.39 | 91.28±0.13 | 71.07±0.26 | 8.67 |
| | Crossformer [ICLR'23] | 75.98±0.15 | 68.46±0.28 | 65.58±0.14 | 66.80±0.10 | 92.13±0.07 | 72.38±0.26 | 7.00 |
| | iTransformer [ICLR'24] | 70.39±0.13 | 61.04±0.30 | 56.71±0.27 | 58.30±0.17 | 87.59±0.15 | 62.17±0.21 | 11.00 |
| | MTST [PMLR, 2024] | 73.97±0.22 | 65.90±0.44 | 62.79±0.68 | 64.05±0.41 | 90.30±0.22 | 68.84±0.26 | 10.00 |
| | Medformer [NeurIPS'24] | 81.60±0.16 | 77.20±0.28 | 73.59±0.19 | 75.17±0.16 | 95.42±0.08 | 82.29±0.15 | 4.00 |
| | SoftShape [ICML'25] | 78.69±0.19 | 72.87±0.51 | 68.52±0.62 | 70.03±0.42 | 93.92±0.10 | 77.18±0.26 | 5.00 |
| | BioFormer (Ours) | 85.73±2.26 | 82.35±3.18 | 80.57±2.99 | 81.38±3.07 | 96.97±0.80 | 88.29±2.98 | 3.00 |

