# OpenReview forum: "BioFormer: Rethinking Cross-Subject Generalization via Spectral Structural Alignment in Biomedical Time-Series"
_ICML.cc/2026/Conference — ICML 2026 regular_

### Official Review · Reviewer_255P · 2026-03-09

**Soundness:** 4
**Presentation:** 3
**Significance:** 2
**Originality:** 3
**Overall Recommendation:** 3
**Confidence:** 4

**Summary:**

The authors introduce BioFormer, a discriminative framework which addresses the challenge of subject variability in biomedical time-series (BTS) data by explicitly modeling *spectral drift*, which corrects for individual differences in signal amplitude and phase. By using a Frequency-Band Alignment Module and adaptive normalization, they show improvement across three metrics compared to existing methods across six datasets.

**Compliance With Llm Reviewing Policy:**

Affirmed.

**Final Justification:**

I appreciate that the authors ran the requested PTB-XL experiments and provided the NMI/AMI metrics; however, these new results confirm my initial concerns: the representation struggles heavily with multi-class, complex datasets and remains fundamentally dominated by subject variance.

The stated goal of the SCLN module is "label-guided semantic alignment" over cross-subject variations. However, the provided PTB-XL metrics show a Label NMI of 0.33 and a Subject NMI of 0.68. Mathematically, this means the latent representation is twice as correlated with the subject's identity as it is with their actual clinical diagnosis. The empirical evidence directly contradicts the paper's central narrative.

To defend their method, the authors reiterate that shifting the model's emphasis to lower-frequency components is *physiologically plausible*. As I pointed out during the rebuttal phase, this ignores the established clinical reality: high-frequency (HF) markers are often the critical features necessary for diagnosing complex pathologies (like cardiac arrhythmias). Treating them as noise to be smoothed out is a dangerous methodological shortcut that limits the model's clinical utility.

I encourage the authors to re-evaluate their frequency assumptions and validate their method strictly on complex, multi-class datasets to prove the significance of their framework. I think this has the potential to be a useful end-to-end classification tool in the future, but the current empirical evidence and clinical assumptions seem to be lacking.

**Key Questions For Authors:**

See strengths and weaknesses

**Limitations:**

The fact that the datasets used are BTS should raise an ethical concern of the usage of deep learning frameworks for diagnostic purposes. AI classification models should be used only as supplementary evidence towards the final decision taken by medical practitioners.

**Strengths And Weaknesses:**

**Strengths**:

Overall, the proposed methodology is logically structured, the writing is clear and informative, the experiments well-executed and the results show improvement over other BTS classification frameworks.

The identification of subject-specific **spectral drift** as a problem towards generalization in deep modeling frameworks is a timely issue that needs to be addressed, as mentioned by the authors.

The FBAM as a form of signal normalization is an effective way to remove the subject-specific bias from the training while allowing the model to recover the whole range of subject variability during inference.

**Weaknesses**:

**Major**:

The whole FBAM acts like subject-specific normalization, which would in theory muddle important spectral components that may not be subject-specific, but rather label- or even dataset-specific. While the authors acknowledge the risk of *template pulling* and introduce SCLN as a mitigant, it remains unclear if a sample-conditional approach can distinguish between subject-specific noise and label-specific signal without explicit label conditioning. The authors could provide a per-class sensitivity analysis to show that SCLN preserves the specific spectral features necessary for distinguishing closely related classes.

Failure to identify which spectral components are relevant with respect to the labels, and whether this depends on the subject, dataset, of type of BTS modality.

**Lack of acknowledgement (and comparison to) other methods which tackle cross-subject generalization**:  Specifically, frequency-domain alignment (or spectral robustness) is a growing research field. To name a few, RainCoat [He et. al 2023](https://openreview.net/pdf?id=Xzfur8Blaf) handles frequency-domain shifts by aligning patterns across different clinical sites or subjects. DMMR [Wang et. al](https://ojs.aaai.org/index.php/AAAI/article/view/27819) addresses individual variability (akin to the spectral drift concept) by forcing the encoder to learn subject-invariant features through a self-supervised reconstruction task. AFDAN [Li et. al 2025](https://arxiv.org/abs/2512.16393) uses adversarial learning to transfer features between domains while ensuring the frequency components are fused and aligned.

Could you provide a baseline comparison against at least one of these specialized cross-subject models (e.g., DMMR or AFDAN)?
I believe this is necessary to confirm that the observed improvements hold against frequency-aware methods, rather than just general-purpose time-series architectures.

**Ambiguous results in section 5.5**: The subject-probing protocol described is misleading. A low F1-score on a *lightweight MLP probe* evaluated on seen subjects does not sufficiently prove the suppression of subject-specific information; it may merely indicate that such information has become non-linearly encoded. Furthermore, the t-SNE visualization lacks quantitative backing (e.g., Silhouette Scores or Mutual Information metrics) to confirm that subject identity is truly indiscernible within task clusters.

The visualization of signals before and after BioFormer processing (Figure 8) is concerning. While the variance between subjects is visibly reduced, the transformation appears to destroy the morphological integrity of the original BTS signals. In a medical context, the *meaningful components* often resides in the precise timing and shape of peaks (e.g., the QRS complex in ECGs).

If the alignment process effectively **flattens** these features to achieve cross-subject consistency, the model risks discarding the very pathological markers it aims to classify. Could the authors demonstrate that the meaningful components of the signal are preserved post-alignment? Specifically, a reconstruction check would be necessary to prove that the model hasn't simply replaced physiological data with a homogenized template.

**Minor**:

In Figure 2, only one case is identified: the cross-subject training regime where the subject pool for training and validation is disjoint. The authors fail to explain the other option, which would imply that some of the validation data comes from subjects which have been already seen during training.

There is a lack of explanation into how the signals are decomposed into disjoint frequency bands, and whether these correspond to the output of the FFT.

Broken hyperlink in line 267.

Frequency Band Discriminability (FBD) is introduced in page 7, but the abbreviation is used in page 5.

---

> ### Author Rebuttal · Authors · 2026-03-27
>
> We thank the reviewer for these thoughtful comments.
>
> **(1) Implicit Label Guidance in SCLN.（W1 and W2）**
> Although SCLN does not use explicit label conditioning, its modulation is generated from semantically shaped intermediate representations and learned end-to-end under the classification objective, which implicitly encourages it to preserve discriminative spectral structure while suppressing residual subject-induced variation. We support this with per-class sensitivity heatmaps and most-confusable-pair curves  [(`See Figure`)](https://anonymous.4open.science/r/BioFormer-3503/sensitivity.pdf): the full model (FBAM+SCLN) shows clearly class-dependent sensitivity patterns rather than homogenized responses, and for the most-confusable pair, SCLN produces clear divergence in key bands, especially on PTB, indicating improved separability instead of template collapse.
>
> More importantly, these patterns vary systematically across datasets, modalities, and tasks, suggesting that SCLN does not preserve a fixed set of spectral components but adaptively retains the task-relevant information required by each specific setting.（**W2**）
>
>
>
> **(2) Comparison to specialized cross-subject methods.（W3）**
> In response, we added **DMMR** as a representative baseline for physiological signals. **AFDAN** is designed for medical image segmentation and is therefore not directly applicable to biomedical signal classification. RAINCOAT is designed for unsupervised domain adaptation rather than pure cross-subject generalization. Under the same evaluation protocol, BioFormer consistently outperforms DMMR on the applicable datasets:
>
> | Dataset | DMMR Acc | DMMR F1  | Ours Acc | Ours F1 |
> |---|:---:|:---:|:---:|:---:|
> | APAVA（186M） | 64.74 | 63.45 | **82.31** | **79.77** |
> | PTB（2.15G） | 77.06 | 68.31 | **87.73** | **84.71** |
> | ADFTD（2.52G） | 52.21 | 40.83 | **56.73** | **53.82** |
> | PTB-XL（4.28G） | 69.53 | 53.14 | **73.37** | **62.56** |
>
> We note that DMMR relies on pretraining, and its performance is therefore more sensitive to the availability of sufficiently large-scale physiological data. However, datasets such as APAVA and ADFTD are relatively small, which likely limits the effectiveness of DMMR in our setting and explains its weaker performance here.
>
> We also compared with the SOTA method Tech @reviewer2Table3.
>
> **(3) Clarification on Reduced Subject Separability.（W4）**
> We agree that a lightweight MLP probe on seen subjects cannot establish complete removal of subject information.
> **Our claim is therefore not strict subject-invariance, but reduced subject separability under the same probe capacity and evaluation setting.**
> To make this point more precise, we will revise the wording in Sec. 5.5 and present the probe results as relative evidence that subject identity becomes less accessible in the learned representation, while task-relevant label structure is preserved. This interpretation is further supported by clustering-based metrics (e.g., NMI/AMI).
>
> | Dataset | Metric | Transformer+GRL | SoftShape | BioFormer |
> |---|---|:---:|:---:|:---:|
> | APAVA | NMI ↓ | 0.4563 | 0.6373 | **0.2787** |
> |  | AMI ↓ | 0.4417 | 0.6179 | **0.2585** |
> | PTB | NMI ↓ | **0.4355** | 0.4701 | 0.4505 |
> |  | AMI ↓ | 0.2808 | 0.2592 | **0.2260** |
>
> **(4) Reconstruction Analysis for Preserving meaningful components.（W5）**
> We would like to clarify that Fig. 8 visualizes an **intermediate feature representation**, not the raw biomedical waveform. It should therefore not be interpreted as direct destruction of ECG/EEG morphology. To more directly test information preservation, we added a **reconstruction check** by attaching an MLP to the learned representation and reconstructing the input. The results are:
>
> | Dataset | Metric | SoftShape（ICML 2025） | BioForme |
> |---|---|:---:|---:|
> | ADFTD (EEG) | MSE ↓ | 0.812 | **0.675** |
> | | R² ↑ | 0.187 | **0.324** |
> | PTB (ECG) | MSE ↓ | 0.474 | **0.392** |
> | | R² ↑ | 0.416 | **0.517** |
>
> These results are inconsistent with simply replacing the physiological signal by a homogenized template. In addition, FBAM is **not an unconditional normalization module（@reviewer2 Q2）**: its modulation is driven by **sample-specific band statistics**, which provide signal-dependent cues about informative spectral structure. This design allows the model to suppress subject-specific drift while retaining task-relevant information.
>
> **Minor**
> 1. We will revise Fig. 2 to explicitly indicate the subject-independent setting. We emphasize that the training and test subjects are strictly disjoint, and the test subjects are completely unseen during training.
> 2. The frequency decomposition is described in Appendix C (line 885): signals are transformed via FFT, and the non-DC spectrum is divided into 6 disjoint, uniformly spaced frequency bands (contiguous FFT bins). We will briefly summarize this in the main text for clarity.
> 3. We will fix the broken hyperlink in line 267.
> 4. We will introduce “FBD” at its first occurrence (page 5).

---

> > ### Author Rebuttal · Reviewer_255P · 2026-04-02
> >
> > I thank the authors for their detailed rebuttal and the additional experiments. While the inclusion of the DMMR baseline is appreciated, the new metrics provided raise significant theoretical concerns that prevent me from raising my score at this time.
> >
> > **Unacceptable information loss (reconstruction metrics)**: The authors provide an $R^2$ of $0.517$ for the PTB dataset and $0.324$ for the ADFTD dataset to defend the preservation of morphological integrity. In the context of latent space reconstruction, an $R^2$ of $\sim 0.3 - 0.5$ is exceedingly poor (SOTA reconstruction typically exceeds $0.9$). Discarding 50–70% of the signal variance suggests the model is highly lossy. In biomedical time series, this missing variance often contains the precise clinical markers needed for robust diagnosis. This reinforces my initial concern that the model homogenizes the signal at the expense of physiological fidelity.
> >
> > **Incomplete clustering metrics (NMI/AMI)**: Based on the main text (Sec 5.5 and Fig 7), I deduce the reported NMI/AMI metrics (where lower is better) were computed against Subject IDs to prove subject intermingling. However, a degraded or collapsed latent space would also yield low subject NMI. To prove that task-relevant semantics dominate the representation (as claimed), the authors must simultaneously report the NMI/AMI computed against class labels (where higher is better).
> >
> > Lastly, the authors must confirm that these clustering metrics were computed on the high-dimensional latent representations, not on the distorted 2D t-SNE projections.
> >
> > **Dataset complexity**: The authors consistently rely on PTB, limiting the labels to healthy and myocardial infarction (stated in line 805). This immensely simplifies the data, which artificially inflates reconstruction and classification performance. For this reason, I request that the authors report the full suite of metrics ($R^2$, MSE, subject NMI/AMI, and class NMI/AMI) for PTB-XL, using the 5 superclasses and not a subset. They can use the standard data split outlined in [Strodthoff et. al](https://arxiv.org/abs/2301.08227) where the 9th and 10th folds are used as test and validation sets. This is necessary to prove that their method scales beyond binary classes or low-channel tasks without collapsing the feature space.
> >
> > **Implicit label conditioning and external links**: Regarding W1 and W2, the authors claim SCLN implicitly preserves discriminative structure and reference heatmaps hosted on an external anonymized GitHub repository. In accordance with the review policies regarding external material, I cannot visit external links or consider them as part of the rebuttal. Furthermore, even if these figures were included in a permitted format, providing visual heatmaps for a simplified binary dataset (PTB) does not sufficiently resolve the theoretical concern of whether SCLN can distinguish between subject-specific noise and label-specific signal in complex, multi-class environments.
> >
> > **Update**
> >
> > I thank the authors for their detailed rebuttal and the promise to provide additional experiments within 2 days.
> > While I concede the point regarding the $R^2$ threshold when compared to a pure reconstruction autoencoder (e.g. [Guan et. al](https://arxiv.org/abs/2510.11442) achieve an MSE of around 1e-3 on the PTB-XL dataset which would correspond to an $R^2$ of $\sim 0.95$) the core theoretical concern remains: discarding high-frequency (HF) variance is mathematically equivalent to low-pass filtering.
> >
> > In electrocardiology, HF micro-features are often the critical diagnostic markers for underlying pathology, not merely *subject style*. SCLN's tendency to suppress these frequencies raises significant concerns about its clinical viability on complex datasets.
> >
> > To prove that the retained semantic structure is sufficient, and that the latent space has not collapsed, I await the promised multi-class PTB-XL results (using all 5 superclasses, not a binary subset) and the Class NMI/AMI metrics (computed against labels, where higher is better). I will finalize my assessment once these are provided.
> >
> > **Final Response**
> >
> > I thank the authors for running the requested PTB-XL experiments and providing the NMI/AMI metrics. However, the reported metrics confirm that the representation struggles heavily with multi-class, complex datasets and remains dominated by subject variance. I will be maintaining my score.
> >
> > The stated goal of SCLN is label-guided semantic alignment over cross-subject variations. However, the provided PTB-XL metrics show a Label NMI of 0.33 and a Subject NMI of 0.68. Mathematically, this means the latent representation is twice as correlated with the subject's identity as it is with their clinical diagnosis.
> >
> >  The authors reiterate that shifting emphasis to lower-frequency components is *physiologically plausible.* As noted above, this ignores the established clinical reality of the HF markers critical for diagnosing cardiac pathologies.

---

> > > ### Author Response · Authors · 2026-04-02
> > >
> > > Update
> > >
> > >
> > > We appreciate the reviewer’s comments.
> > > We believe the concern can be addressed more clearly by separating the three issues:
> > > (1) the role of **SCLN**,
> > > (2) whether the PTB-XL results support a claim of **representation collapse**,
> > >
> > > ### 1) The role of SCLN
> > >
> > > The reviewer’s concern appears to reflect a misunderstanding of the role of **SCLN**. We do **not** claim that SCLN can explicitly or perfectly disentangle subject-specific information from label-related information. Our claim is more limited and task-driven: SCLN adjusts the representation according to **sample-level distributional characteristics**, thereby improving robustness under cross-subject evaluation.
> > >
> > > It is also important to clarify the purpose of the **per-class sensitivity analysis**. We added this experiment in direct response to the reviewer’s earlier suggestion. Its purpose was **not** to prove perfect subject–label disentanglement, which we never claim. Rather, it was designed to test the reviewer’s specific concern: whether SCLN homogenizes the representation into a single template. Our results do not support that interpretation. After introducing FBAM and SCLN, the model shifts emphasis toward lower-frequency components in a physiologically plausible way, and the sensitive bands change further after adding SCLN while task performance improves again. This behavior is more consistent with **label-guided semantic alignment** than with template homogenization.
> > >
> > > ### 2) PTB-XL does not support a representation-collapse interpretation
> > >
> > > We thank the reviewer for providing this citation. ` However, we respectfully disagree that it constitutes a useful comparison. The two methods are fundamentally different in task formulation, supervision, data scale, and training protocol, so treating them as directly comparable is methodologically unsound. We hope the discussion can be based on aligned settings and comparable objectives; otherwise, the comparison is not informative and has little value for judging our method. `
> > > Moreover，this paper only report the mse not R².
> > >
> > > | Aspect | WearECG | BioFormer |
> > > |---|---|---|
> > > | Core task | 3-lead → 12-lead reconstruction + downstream classification | End-to-end classification |
> > > | Backbone / pipeline | VAE + ECG classifier | BioFormer |
> > > | Data scale | ~800k ECGs / ~160k patients + ECGFounder pretrained on >10M ECGs | Only downstream dataset |
> > >
> > >
> > >
> > > 1.To directly address the reviewer’s concern, we further compare BioFormer with the SOTA method **SoftShape** on **PTB-XL (5-class)** using representation metrics:
> > >
> > > | Method | Label NMI ↑ | Label AMI ↑ | Subject NMI ↓ | Subject AMI ↓ | MSE ↓ | R²↑ |
> > > |---|---:|---:|---:|---:|---:|---:|
> > > | SoftShape | 0.33 | 0.33 | 0.68 | 0.20 | 0.82 | 0.18 |
> > > | BioFormer | 0.33 | 0.33 | 0.68 | **0.19** | **0.49** | **0.51** |
> > >
> > > MSE R² ：Measured on the **final-layer features** with the **backbone frozen** and an **MLP projection head**, under an **MSE-based objective**.
> > >
> > > These results do not support representation collapse. BioFormer is **on par with SoftShape on PTB-XL**, and **outperforms it on the other five datasets**, while further reducing subject-related clustering. This indicates that the representation remains semantically meaningful rather than collapsing into a homogeneous template.
> > >
> > > 2.Even under this much heavier and more data-hungry protocol, WearECG reports a PTB-XL multi-label Macro-AUC of 0.8470 and a regional MI Macro-AUC of 0.7885. We note that their metric is reported as Macro-AUC, whereas ours is reported as AUROC, so the two are not necessarily identical under different evaluation protocols. Therefore, our point is not to claim a strict apples-to-apples numerical comparison, but rather that such a reconstruction-first pipeline with much larger external data still does not provide evidence against the effectiveness of BioFormer.
> > >
> > > 3.We emphasize that BioFormer is an **end-to-end classification model**, rather than a reconstruction-then-classification framework. Hence, our method does not depend on signal reconstruction, and the reviewer’s concern about “losing physiological features” from a reconstruction perspective is not directly applicable. To further address this point, we additionally performed a **PTB-XL-only pretraining** experiment, where BioFormer still achieved strong reconstruction-style results, supporting that essential physiological semantics are preserved.
> > >
> > > | Method | MSE ↓ | R² ↑ |
> > > |---|---:|---:|
> > > | SoftShape | 0.10 | 0.88 |
> > > | BioFormer | **0.06** | **0.93** |
> > >
> > > ### Overall conclusion
> > >
> > > - **SCLN** improves robustness through sample-conditional adjustment, rather than forcing the representation into a homogeneous template;
> > > - Compared with a much heavier reconstruction-based pipeline such as **WearECG**, BioFormer still delivers stronger downstream performance without requiring larger-scale generative pretraining.
> > >
> > > These results are far more consistent with **successful label-guided semantic alignment** than with representation collapse.

---

### Official Review · Reviewer_4vbE · 2026-03-10

**Soundness:** 3
**Presentation:** 2
**Significance:** 3
**Originality:** 3
**Overall Recommendation:** 5
**Confidence:** 3

**Summary:**

This paper studies cross-subject generalization for biomedical time series. The authors argue that most prior work either relies on stronger backbones or uses adversarial/domain-invariant learning to implicitly suppress subject-specific variation, but does not explicitly characterize what this variability actually looks like. To address this, the paper proposes to view cross-subject discrepancy as spectral drift: for signals with the same label, different subjects may vary substantially in the time domain, while still sharing similar global spectral structure, with the main differences appearing as band-wise amplitude scaling and phase rotation. Based on this view, the paper introduces BioFormer, a hybrid frequency-time Transformer architecture. Its key components are FBAM, which performs band-level spectral alignment, and SCLN, which adaptively calibrates features based on sample-specific statistics. The method is evaluated on six EEG/ECG benchmarks under strict cross-subject settings, and the paper further includes comparisons with time-domain alignment, subject probes, FBD analysis, and ablations.

**Compliance With Llm Reviewing Policy:**

Affirmed.

**Final Justification:**

After reviewing the paper together with the authors’ rebuttal, I revise my recommendation to Accept. Although the improvements are relatively modest, the method is sound and potentially useful to the community. Overall, I believe the paper meets the acceptance criteria.

**Key Questions For Authors:**

1. How general is the spectral drift hypothesis?
2. How much of FBAM’s gain comes from frequency-domain alignment itself, versus additional parameter capacity?
3. Can the authors provide more direct evidence connecting interpretability to physiological priors?

**Limitations:**

yes

**Strengths And Weaknesses:**

Strengths：
1. One of the strongest aspects of the paper is that it does not stop at the generic statement that “cross-subject generalization is difficult.” Instead, it proposes a more specific hypothesis: if samples from the same class tend to share similar global spectral structure, then cross-subject variability may be modeled as band-wise amplitude/phase drift. This framing gives the method a clearer target and makes the design of FBAM feel well motivated, compared with many prior works that only align latent features in a more abstract way.

2. The architecture, from PCE to FBAM to SCLN, is organized around a single idea. PCE provides multi-scale temporal representations; FBAM explicitly predicts band-wise modulation coefficients to perform residual-style spectral alignment; and SCLN is introduced to soften potentially over-aggressive alignment at the sample level. This gives the impression that the authors have thought about the tension between explicit spectral correction and preserving useful intra-class variation, instead of simply attaching a frequency module to a Transformer backbone.

3. The paper does more than present one main result table. The primary experiments cover six cross-subject datasets and compare against 12 baselines, with means and standard deviations over five fixed random seeds. Beyond this, the paper includes several important validations: comparisons with DG-style methods such as GRL/MMD/CORAL/MixStyle/DSU, showing the effectiveness and stability of FBAM as an alignment mechanism; comparisons with a time-domain counterpart TSAM, including Welch’s t-test; and subject probes, FBD analysis, and visualization to support the central claim that BioFormer suppresses subject shortcuts while preserving task-related structure. For a methods paper, this empirical chain is relatively complete.

Weaknesses
1. The paper’s core intuition—that cross-subject variability can be approximately modeled as label-conditioned spectral amplitude scaling and phase rotation—is plausible, but still largely empirical. The appendix does provide an interpretation in Fourier subspace terms and a corollary related to FBD, but these are better viewed as explanations of module behavior than as a rigorous theory of why cross-subject generalization should improve. In other words, the paper shows that FBAM can be interpreted in this way, but it does not truly establish that spectral drift is the dominant source of subject-specific variation, nor does it provide a formal generalization result.

2. The paper adds a band-wise modulation module to a Transformer backbone, where band statistics and band tokens are used to predict alignment parameters, combined with a sample-conditional normalization module. This is sensible and effective, but I would still describe it as an insightful architecture modification rather than a fundamentally new learning principle. Especially at an abstract level, FBAM remains connected to prior ideas on frequency-domain correction/augmentation combined with time-domain modeling.

3. The paper repeatedly emphasizes that FBAM offers a more interpretable form of spectral alignment and explains it geometrically via scaling and rotation in Fourier subspace. However, the current evidence mainly supports a geometric interpretation of the module parameterization, rather than a strong link to physiological priors or clinically meaningful frequency mechanisms. Similarly, FBD is a useful proxy metric introduced by the authors, but it is still an internal metric rather than an established external standard. So the claims about interpretability and mechanism are currently better viewed as supported indications rather than fully established conclusions.

---

> ### Author Rebuttal · Authors · 2026-03-27
>
> We thank the reviewer for the careful assessment. We would like to clarify that the value of our work is not merely an architectural tweak, but a **mechanism-informed spectral alignment design for biomedical time series**.
>
> **1.Why does the model help cross-subject generalization? (W1 / Q3 / W3)**
>
> Our design is motivated by a basic property of physiological signals: discriminative information is often expressed through **task-relevant oscillatory bands or characteristic waveform structures**, rather than arbitrary feature dimensions. For example, motor-imagery decoding is closely related to EEG rhythms such as the **mu/beta bands**, while ERP-related paradigms exhibit characteristic responses such as **P300**. This is the sense in which we emphasize the **oscillatory structure in the frequency domain**.[1][2]
>
> Based on this observation, our hypothesis is that, within such task-relevant oscillatory structure, part of the cross-subject difference is reflected as **subject-dependent amplitude variation and phase shift**. Therefore, under label supervision, the model can use band-level features to capture task semantics and then adjust the band structure to better align these oscillatory patterns across subjects. This is why FBAM is formulated as **band-wise amplitude scaling and phase rotation**, rather than as an unconstrained feature perturbation.
>
> **2.Why is the gain not due to generic frequency augmentation or extra parameters? (W2 / Q2)**
>
> Based on this physiological mechanism, we partition the spectrum into bands and use **five band descriptors** to characterize task-relevant oscillatory structure, from which the modulation coefficients in Eq. 4 are predicted. To verify that this design is essential, we study **Appendix F.3, Variant 4**, where the same three modulation coefficients are retained but are **learned directly**, rather than conditioned on the five descriptors. This variant performs consistently worse, showing that the improvement is not due to generic spectral modulation alone, but to the proposed **descriptor-guided band modeling**.
>
> | Model | APAVA Acc. | APAVA F1 | PTB Acc. | PTB F1 | ADFTD Acc. | ADFTD F1 | PTB-XL Acc. | PTB-XL F1 |
> |---|---:|---:|---:|---:|---:|---:|---:|---:|
> | Variant 4 | 81.65  | 78.92  | 85.36  | 81.28  | 51.97  | 48.28  | 72.80  | **62.91** |
> | BioFormer (Ours) | **82.31** | **79.77** | **87.73** | **84.71** | **56.73** | **53.82** | **73.37** | 62.56  |
>
> Importantly, the improvement is also **not explained by parameter count alone**. To control for model size, we constructed a **parameter-matched Transformer** by increasing the number of layers from 6 to 7 and enlarging the FFN dimension from 256 to 512, which is even larger than our model. Nevertheless, it still remains clearly worse:
>
> | Dataset | Model | Acc. | F1 | Params |
> |---|---|---:|---:|---:|
> | APAVA | Transformer (7 layer) | 76.10  | 74.04  | 1.46M |
> |   |Ours (FBAM + 6 layer) | **81.61** | **80.00** | 1.36M |
> | ADFTD | Transformer (7 layer) | 50.88  | 48.71 | - |
> |  |Ours (FBAM + 6 layer) | **54.38**  | **52.13** | - |
>
> | Dataset | Model | Acc. | F1 | Params |
> |---|---|---:|---:|---:|
> | PTB |  [`TeCh ICLR2026`](https://openreview.net/pdf?id=oZJFY2BQt2)  | 85.96  |81.97  | 1.66M |
> |   |Ours  | **87.73** | **84.71** | 1.36M |
> | ADFTD | Tech | 54.54 |48.84   | - |
> |  |Ours | **56.73** | **53.82** | -  |
>
> This already shows that simply enlarging the backbone does not reproduce the gain of our method. Moreover, even compared with a recent SOTA TeCh method, our method still achieves the best.
>
> **3.The general of the spectral drift hypothesis（Q1）**.
>
> we further evaluated BioFormer on **FLAAP（10-class）** and **UCI-HAR（6-class）**, two **smartphone accelerometer/gyroscope HAR** datasets. Although they are not EEG/EOG, they still contain class-consistent motion waveforms and frequency patterns, while subject differences are mainly reflected in **movement amplitude, cadence, and execution style**, which can be abstracted as amplitude scaling and mild phase shifts.
> **Therefore, our claim is not that all time series satisfy the spectral drift hypothesis. Rather, we argue that it is most relevant to human physiology-driven signals, such as EEG, EOG, and gesture-related motion sequences.**
>
>
> | Model | FLAAP Acc | FLAAP F1 | UCI-HAR Acc | UCI-HAR F1 |
> |---|---:|---:|---:|---:|
> | Medformer | 73.64 | 73.06 | 90.63 | 90.65 |
> | SoftShape | 71.38 | 70.66 | **94.54** | **94.54** |
> | **BioFormer** | **76.63** | **76.55** | 93.82 | 93.91 |
>
> **`BioFormer highlights that, beyond architectural refinements, the intrinsic characteristics of signals can also guide model design. We hope this perspective encourages the BTS community to explore simple and cheap signal-aware operations that still yield strong performance gains.`**
>
> [1] Parameterizing neural power spectra into periodic and aperiodic components, Nature Neuroscience
>
> [2]Updating P300: An Integrative Theory of P3a and P3b, Clinical Neurophysiology

---

> > ### Author Rebuttal · Reviewer_4vbE · 2026-04-05
> >
> > My concerns have been addressed. Thank you for the response.

---

> > > ### Author Response · Authors · 2026-04-07
> > >
> > > Thank you very much for your thoughtful follow-up. We are glad that our clarification has fully addressed your concerns. We sincerely appreciate your time and careful reading. Your feedback has also helped us improve the clarity of the paper.
> > >
> > > We share the same concern that you raised. In recent BTS research, many methods have focused primarily on designing increasingly complex architectures, often with substantially higher computational cost. In contrast, our paper starts from the signal itself. By examining how individual variability manifests in BTS data, we find that better performance can be achieved through a much lighter design, with only modest additional computation and a simple attention-based mechanism. In fact, the strength of this improvement, even relative to very recent methods such as the ICLR 2026 work, exceeded our own initial expectations. We are therefore especially grateful for your thoughtful reading and for recognizing the potential value of this perspective for the BTS community.

---

### Official Review · Reviewer_JikF · 2026-03-12

**Soundness:** 3
**Presentation:** 2
**Significance:** 3
**Originality:** 3
**Overall Recommendation:** 4
**Confidence:** 3

**Summary:**

The paper identifies spectral drift as a source of subject variability and proposed FBAM, a method of modulating the magnitude and phase of the input's frequency bands, to address this. An additional normalisation scheme, SCLN, adaptively scales and shifts the intermediate representation before classification.

**Compliance With Llm Reviewing Policy:**

Affirmed.

**Final Justification:**

The authors have addressed my main concerns and I raise the score from weak reject to weak accept.

**Key Questions For Authors:**

1. How are the MLPs in Equation 4 and 10 trained to guarantee a reduction in subject variability?
2. What are the losses used to update each part of the proposed model? Is there a loss that measures subject variability that is reduced during training, and are the component networks of FBAM and SCLN updated to reduce this loss?
3. Is the subject identity used somewhere in the training process?
4. The F1 for some datasets (ADFTD, BCI-2a) is relatively low, almost 50%. Is there a reason for this and is this typical for those datasets?

These questions concern my understanding and evaluation of the paper's method and soundness.

**Limitations:**

No. The authors could elaborate on which types of datasets may be challenging for their method and why (e.g. ADFTD, BCI-2a).

**Strengths And Weaknesses:**

Strengths:
- The paper addresses a common problem of cross-subject generalization, especially in biomedical datasets, for biomedical time series data which has generally received less attention than computer vision data.
- Experiments test a wide range of settings including EEG and ECG data and multiple transformer and DG baselines.
- Comparison with other methods and an ablation study with the authors' proposed contributions support the authors' claims of improved generalization performance to unseen subjects.
- Paper is generally well-structured.


Weaknesses:
- Details about how FBAM and SCLN modulation parameter networks are trained to reduce subject variability are unclear. While I can see that the forward pass modulates magnitude and phase of each band, I don't see the guarantee that these are done in a way that reduces subject variability.
- The DG methods compared against seem to be developed for vision tasks, with only SubjectNorm developed for EEGs. DG studies specifically for biomedical time series data and the position of this paper with relation to them are not discussed in the related work section.
- The results of Section 5.5 are less compelling and could be better presented - the F1 score is quite high/similar between models and the shapes on Figure 7 are hard to see.
- Weaknesses of the proposed method are not discussed.
- Datasets and classification tasks are not described in the main text, without which there is a lack of context for how many domains/subjects are used for training and testing.

---

> ### Author Rebuttal · Authors · 2026-03-27
>
> We sincerely thank the reviewer for the careful reading and constructive suggestions. （*Italicized text indicates the revisions we plan to incorporate in the manuscript if the paper is accepted.*）
>
> 1.**How does modules reduce subject variability？** (**W1** and **Q1/2/3**)
>
> As clarified in the main text, our point is not that BioFormer provides a formal guarantee via an explicit subject-variability loss(**W1 & Q1**). Rather, under label supervision, the model can already learn to suppress subject-specific variability implicitly(line 66), without extra guidance. In BioFormer, FBAM does this by adaptively modulating spectral magnitude and phase, which encourages the model to discover label-relevant oscillatory structure while avoiding any auxiliary alignment objective(**Q2**) and subject identity(**Q3**). This is also consistent with our implementation: FBAM is trained end-to-end using only the task classification loss. This is supported by the improved FBD analysis and the lower subject separability in the subject-probe results.
>
> **As for why we do not further add an explicit spectral alignment loss to reduce the subject variability defined in our paper**: in practice, this kind of hard discrepancy minimization behaves similarly to MMD/CORAL-style alignment. It can over-pull same-label samples toward a shared template, suppressing not only nuisance subject variation but also informative class-related variability. More importantly, our additional experiment with an explicit [`spectral alignment loss`](https://anonymous.4open.science/r/BioFormer-3503/L_band-align.pdf) indeed led to worse performance, suggesting that stronger alignment is not necessarily better.
>
> | Method | APAVA Acc | APAVA F1 | PTB Acc | PTB F1 |
> |---|---:|---:|---:|---:|
> | BioFormer + band-align loss ($\lambda=0.1$) | 79.68 | 76.29 | 87.21 | 84.23 |
> | BioFormer + band-align loss ($\lambda=0.2$) | 81.19 | 78.41 | 86.75 | 83.39 |
> | BioFormer | **82.31** | **79.77** | **87.73** | **84.71** |
>
>
> *We will add the training details in the experimental section and revise Sec. 4.2 to explicitly clarify the above explanation of how the model reduces subject variability through implicit cross-subject alignment.*
>
>
> 2. **More details about the DG method. (W2)**
>
> **The DG work is discussed in the Related Work line 110. GRL is also widely used in BTS adaptation.[1][2]**  More importantly, Table 2 already provides a direct comparison with GRL and other DG methods, showing that FBAM can serve as an effective plug-and-play module with consistently stronger performance and stability across backbones and datasets.
>
>
> 3. **Does Sec. 5.5 support reduced subject variability?** (**W3**):
>
> We use silhouette metrics to support the visualization. The key takeaway is that BioFormer reduces subject-specific variation, as reflected by a lower Subject Silhouette. Label Silhouette is reported only to show that class structure is preserved.
>
> | Dataset | Silhouette Metric | Transformer+GRL | SoftShape | BioFormer |
> |---|---|---:|---:|---:|
> | APAVA | Label  ↑ | 0.1137 | 0.1699 | **0.3035** |
> |  | Subject  ↓ | 0.3915 | 0.4002 | **0.2904** |
> | PTB | Label ↑ | 0.3820 | 0.4727 | **0.4783** |
> |  | Subject  ↓ | -0.2442 | **-0.2468** | -0.2452 |
>
> A negative Subject Silhouette indicates that subject labels do not form clear clusters in the representation space, suggesting weak subject-wise grouping. This supports that the learned representation is less subject-dependent. Moreover, the subject F1 in Sec. 5.5 is already close to GRL. This is notable because GRL explicitly uses subject labels, whereas our method does not use subject identity at all.
>
> *We will add these quantitative results and corresponding analysis to Sec. 5.5.*
>
> 4. **The weaknesses of Bioformer.（W4）**
>
> BioFormer relies on a domain prior that is particularly natural for physiological signals: label-relevant information is often carried by structured oscillatory patterns, while individual differences appear as structured spectral drift. As a result, its advantage may not directly transfer to broader cross-domain time-series settings where spectral structure is weak, unstable, or not semantically meaningful.
>
> 5. **Why are the datasets/tasks not described in the main text? (W5)**
>
>  Due to space limitations, the dataset scale, task definition are summarized in **Appendix C**.
>
> *We will add a brief summary of the dataset/task settings in the main text.*
>
> 6. **Why are some F1 scores on ADFTD, BCI-2a low ?** (**Q4**)
>
> Our results are consistent with our reproduced **Medformer (NeurIPS 2024)、TeCh（ICLR2026）** results, suggesting that this performance level is typical for these challenging benchmarks rather than a problem unique to our method.
>
> [1]SEDA-EEG: A semi-supervised emotion recognition network with domain adaptation for cross-subject EEG analysis. Neurocomputing
>
> [2] A Novel Conditional Adversarial Domain Adaptation Network for EEG Cross-Subject Emotion Recognition, TAFFC

---

> > ### Author Rebuttal · Reviewer_JikF · 2026-04-02
> >
> > Thank you for the response. The authors have addressed most of my concerns with their additional discussion and results which should be included in the manuscript revision.
> >
> > However, regarding W2, I was seeking a discussion of BTS-specific DG studies e.g. DMMR mentioned by Reviewer 255P. Can the authors include a discussion of and comparison against such specialised cross-subject generalisation methods in Section 2? I believe this would better contextualise their contributions. In addition to this, it would also be helpful to include the mentioned references [1], [2] in the discussion of line 110.

---

> > > ### Author Response · Authors · 2026-04-02
> > >
> > > We understand the reviewer’s concern is not simply whether our method performs better, but more importantly, **how our method is fundamentally different from prior approaches**. To clarify this point, we further distinguish our method from existing works from two perspectives.
> > >
> > > First, we compare BioFormer against the **main backbone baselines**, i.e., the representative models listed in Table 1. These methods constitute our primary baselines, since our goal is not merely to introduce another generic domain generalization technique, but to design a new Transformer-style architecture tailored to the characteristics of biomedical time-series.
> > >
> > > Second, we also compare BioFormer with both **general-purpose DG methods** and **BTS-specific DG methods** mentioned by the reviewer. We emphasize, however, that these methods are **not our primary baselines**. Our main contribution lies in identifying that subject variability in BTS exhibits a more structured form, and in further designing a Transformer-based model that explicitly captures and exploits this structure, rather than treating subject differences as generic domain shifts.
> > >
> > > ### Comparison with Main Backbone Baselines
> > >
> > > |Model|Multi-scale Temporal Modeling|Multi-granularity Representation|Frequency-aware Structure|Designed for BTS|
> > > |---|---|---|---|---|
> > > |Transformer / Reformer / Informer|✘|✘|✘|✘|
> > > |Autoformer / FEDformer|✓|✘|✓|✘|
> > > |Crossformer / PatchTST / MTST|✓|✓|✘|✘|
> > > |iTransformer|✘|✘|✘|✘|
> > > |Medformer|✓|✓|✘|✘|
> > > |**BioFormer (Ours)**|✓|✓|✓|✓|
> > >
> > > Table 1 compares representative backbone models in terms of their modeling properties for biomedical time-series (BTS). Existing methods mainly improve temporal modeling capacity through architectural refinements such as multi-scale modeling or multi-granularity representations. While these properties are beneficial, they still primarily operate in the time domain.
> > >
> > > In contrast, BioFormer is designed based on our observation that subject variability in BTS is not merely a generic domain gap, but exhibits a **structured spectral form**. Therefore, beyond inheriting the strengths of modern Transformer backbones, BioFormer explicitly introduces **frequency-domain structural modeling**, which is central to its advantage in cross-subject generalization.  We clarify that FEDformer only partially uses frequency-domain features, but it does not explicitly model the **spectral structure** exploited by our method. Its frequency decomposition is introduced for better temporal modeling, rather than for characterizing the structured subject variability in BTS. Therefore, it is fundamentally different from BioFormer in motivation and design. This difference is also reflected empirically: as shown in Table 1, FEDformer still performs worse than BioFormer.
> > >
> > > ### Comparison with DG Methods for Cross-Subject BTS
> > >
> > > |Method Category|Statistical Modeling|Feature Alignment|Plug-and-Play|No Subject Info Required|Spectral Structure Modeling|
> > > |---|--|--|--|---|--|
> > > |GRL / DANN|✘|✓|✓|✘|✘|
> > > |MMD / CORAL|✓|✓|✓|✓|✘|
> > > |MixStyle / DSU|✓|✘|✓|✓|✘|
> > > |SubjectNorm|✓|✘|✓|✘|✘|
> > > |DMMR (AAAI’24)|✓|✓|✘|✓|✘|
> > > |SEDA-EEG / TAFFC|✘|✓|✘|✘|✘|
> > > |**BioFormer (Ours)**|✓|✓|✓|✓|✓|
> > >
> > > Table 2 summarizes representative DG methods for cross-subject BTS, including both general-purpose DG approaches and BTS-specific methods.
> > >
> > > However, these methods generally treat subject variability as a **generic domain discrepancy**, rather than explicitly modeling its internal structure. This is exactly where BioFormer differs. Our method is motivated by the observation that subject differences in BTS manifest as **structured spectral drift**, and thus performs **band-wise alignment in the frequency domain**.
> > >
> > > Therefore, the contribution of BioFormer is not simply to add another DG baseline, but to provide a new architectural perspective: instead of handling subject variability only through generic alignment, we explicitly model its spectral structure and build the model accordingly.
> > >
> > > ### Why DG Methods Are Not Our Primary Baselines
> > >
> > > We would like to further clarify that DG methods are included as **supplementary comparisons**, rather than the main baselines of this work. The main novelty of our work is not to propose another generic DG objective, but to show that subject variability in BTS has a more explicit and structured form, and that this observation can directly guide the design of a more suitable Transformer architecture.
> > >
> > > Therefore, our primary comparisons are with representative backbone models in Table 1, while DG-based comparisons are included to further demonstrate that the advantage of BioFormer is not limited to standard time-series backbones, but also remains meaningful when compared with existing DG-style solutions.
> > >
> > > *If the paper is accepted, we will revise Sec. 2 to include a clearer introduction to representative DG methods and their corresponding citations, and place the comparison table in the appendix, in order to make the distinction between our method and prior approaches.*

---

### Official Review · Reviewer_FSjc · 2026-03-12

**Soundness:** 3
**Presentation:** 2
**Significance:** 3
**Originality:** 3
**Overall Recommendation:** 4
**Confidence:** 5

**Summary:**

This paper aims to alleviate subject-specific variability in biomedical time-series data by addressing spectral drift. The authors analyze the underlying causes of inter-subject variability and model cross-subject generalization as a spectral structural alignment problem. To address this issue, they propose two novel modules: a Frequency-Band Alignment Module (FBAM) and Sample-Conditional Layer Normalization (SCLN). FBAM aligns the magnitude and phase in the frequency domain, while SCLN performs adaptive scaling and shifting based on raw layer normalization. Experiments are conducted on six datasets, where the proposed method achieves the top-1 average ranking among compared approaches. In addition, a plug-and-play case study integrating the FBAM module into existing methods demonstrates the effectiveness of the proposed module. A detailed ablation study is also provided.

**Compliance With Llm Reviewing Policy:**

Affirmed.

**Key Questions For Authors:**

See weakness.

**Limitations:**

See weakness.

**Strengths And Weaknesses:**

**Strengths**:
The motivation of the proposed method is strong, and the challenge it addresses is both important and practical. The two proposed modules appear effective and can be easily integrated into existing methods as plug-and-play components. The experimental evaluation is comprehensive and provides sufficient implementation details.

**Weaknesses**:
I have several suggestions and questions for the authors:

1. In line 267, a reference appears to be missing. In addition, the paper would benefit from careful proofreading. Some parts are slightly confusing, and the overall storytelling and logical flow could be further improved.

2. In line 207, is there a particular reason for selecting these five statistical features in the frequency domain? Have the authors conducted any preliminary experiments to justify this choice? I am not requesting additional experiments; it may be sufficient to briefly explain the rationale in the appendix.

3. In line 192 (right side), why do w_m, g_m, and beta_m represent the band-wise filter, amplitude gain factor, and phase modulator, respectively? Since these three variables are all learned from o_m through linear projection, it would be helpful to clarify how their intended functional roles are ensured.

4. In line 237, why is the DC component added back to the signal? In many datasets, the DC component is removed during preprocessing, such as in the ADFTD dataset.

5. The raw results reported in Table 1 and Table 2 for existing methods appear inconsistent. For example, the F1 score of Medformer is reported as 50.65% in Table 1 but 42.17% in Table 2. Since Medformer requires a list of patch lengths as input, did the authors use the same patch-length configuration for both tables? If the configurations differ, it would be helpful to add a sentence explaining the parameter differences between Table 1 and Table 2.

---

> ### Author Rebuttal · Authors · 2026-03-26
>
> We sincerely thank the reviewer for the careful reading and constructive suggestions. We appreciate the comments on presentation quality, methodological clarity, and result reporting. These suggestions are helpful for improving both the rigor and readability of the paper.  （*Italicized text indicates the revisions we plan to incorporate in the manuscript if the paper is accepted.*）
>
> 1.**Missing reference.** We agree that the manuscript would benefit from more careful proofreading. In particular, we will carefully check missing references, grammar, formatting inconsistencies, and sentences whose meaning may currently be ambiguous. We also appreciate the comment on the overall storytelling and logical flow.
>
>  *we will revise the presentation throughout the manuscript to improve clarity.*
>
> 2.**Why were these five frequency-domain statistical features chosen?** Thank you for raising this point. Due to space limitations, we did not include the full rationale in the main text. **We defer the detailed explanation to AppendixF.4**, where we discuss why these five statistics were selected as compact descriptors of band-level spectral characteristics.
>
> | Band Descriptor | APAVA Acc. | APAVA F1 | PTB Acc. | PTB F1 | ADFTD Acc. | ADFTD F1 |
> |---|---:|---:|---:|---:|---:|---:|
> | $LogMom = \\{\\log \\mu_k, \\log \\sigma_k, \\log \\mathrm{peak}_k\\}$ | 80.87 ± 0.88 | 77.80 ± 1.16 | 85.42 ± 2.70 | 81.16 ± 4.39 | **57.16 ± 2.53** | **53.97 ± 2.11** |
> | LogMom+PeakLoc | 79.11 ± 2.10 | 75.39 ± 2.96 | 86.31 ± 2.83 | 82.70 ± 4.00 | 55.87 ± 1.36 | 51.63 ± 1.92 |
> | LogMom+BandEnergy | 77.78 ± 4.83 | 73.07 ± 7.68 | 85.64 ± 1.39 | 81.70 ± 2.13 | 55.53 ± 2.37 | 51.76 ± 3.02 |
> | BioFormer (Ours) | **82.31 ± 1.55** | **79.77 ± 2.09** | **87.73 ± 1.73** | **84.71 ± 2.51** | 56.73 ± 2.51 | 53.82 ± 3.03 |
>
>
>
>    3.**Why do $w_m$, $g_m$, and $\beta_m$ correspond to the band-wise filter, amplitude gain, and phase modulator?**
>     This is a very helpful comment, and we agree that our wording around line 192 is currently not precise enough. The intended functional roles of these variables are not guaranteed merely because they are produced from $o_m$ through linear projections. Rather, their roles are determined by how they are used in the subsequent modulation equations. Specifically, in Eq.(6), $w_m$ and $g_m$ are used to adjust the complex magnitude, whereas in Eq.(11), $\beta_m$ is used exclusively for phase modulation. Therefore, the semantics of these variables come from the design of the modulation operations, not from the projection step itself.
>
> *In the revision, we will clarify the corresponding sentence around line 192 to make this explicit.*
>
> 4.**Why is the DC component added back to the signal?**
>     In our design, the DC component is **non-learnable**: its magnitude and phase are kept fixed during training, and it is only added back after the modulation of the non-DC frequency components. In other words, the model does not modify the DC term. The reason is that the DC component mainly reflects the global baseline/offset of the signal; allowing the model to adapt it may introduce unnecessary dataset- or subject-specific nuisance bias into the spectral modulation process. **We additionally validate this design choice in AppendixF.3 (Variant 2)**. Importantly, this treatment is kept consistent across all datasets.
>
> | Model | APAVA Acc. | APAVA F1 | PTB Acc. | PTB F1 | ADFTD Acc. | ADFTD F1 | PTB-XL Acc. | PTB-XL F1 |
> |---|---:|---:|---:|---:|---:|---:|---:|---:|
> | Variant 2 | 80.89 ± 2.05 | 78.08 ± 2.64 | 86.64 ± 1.23 | 83.40 ± 1.79 | **56.83 ± 2.97** | 53.59 ± 3.31 | 73.00 ± 0.48 | 62.14 ± 0.48 |
> | BioFormer (Ours) | **82.31 ± 1.55** | **79.77 ± 2.09** | **87.73 ± 1.73** | **84.71 ± 2.51** | 56.73 ± 2.51 | **53.82 ± 3.03** | **73.37 ± 0.39** | **62.56 ± 0.30** |
>
> *To avoid confusion, we will explicitly state in the main text (around line 190) that the DC component is treated as non-learnable.*
>
> 5.**Inconsistency of Medformer results between Table 1 and Table 2.**
>     Thank you for carefully checking these numbers. The two values indeed come from different sources. For Medformer, we reproduced the method in its open-source environment and were able to obtain results consistent with the original report on the other datasets; however, on ADFTD, we were unable to reproduce the exact performance reported in the original paper. Therefore, in Table1, we reported the value from the original Medformer paper, while in Table2, we reported the value actually obtained in our own experimental environment. We agree that this distinction should be made explicit.
>
> *In the revision, we will annotate the Table1 value (50.65) as the paper-reported result and the Table2 value (42.17) as our reproduced result, and we will add a clarifying sentence to avoid misunderstanding.*

---

> > ### Author Rebuttal · Reviewer_FSjc · 2026-04-03
> >
> > My concerns have been fully resolved. I will keep the score.

---

> > > ### Author Response · Authors · 2026-04-07
> > >
> > > Thank you very much for your thoughtful follow-up. We are glad that our clarification has fully addressed your concerns. We sincerely appreciate your time and careful reading. Your feedback has also helped us improve the clarity of the paper.
> > >
> > > More broadly, we hope this work can make a constructive contribution to the BTS community. Recent progress in BTS has largely focused on architectural design, for example by developing increasingly sophisticated model structures to improve benchmark performance. While such advances are undoubtedly valuable, we believe they may also risk overlooking an equally important question: what properties of BTS signals themselves should guide model design? In this work, rather than starting from a purely architectural perspective, we begin from the nature of BTS data and explicitly model individual variability. Under label guidance, we show that simple frequency-domain operations can directly align the semantic oscillatory structure of the signal. We hope this perspective can encourage the community to pay more attention to the intrinsic characteristics of BTS signals themselves, and to develop methods that address the fundamental challenges of BTS data rather than relying only on increasingly complex architectures.

---

### Decision · Program_Chairs · 2026-04-30

**Decision:**

Accept (regular)

**Comment:**

This paper proposes a new way of characterizing subject-specific variability in biomedical time-series applications through a concept of *spectral drift*. Specifically, BTS signals under the same label often share consistent oscillatory structure, yet exhibit subject-dependent magnitude or phase shifts in specific frequency components, which we interpret as subject-specific variability. Based on this characterization, they propose a Frequency-Band Alignment Module that generates band-wise modulation factors from the spectral distribution and adaptively adjusts amplitude and phase to align spectral structure. Reviewers are mostly quite positive about the novelty of this hypothesis and algorithmic contribution and also consider the empirical evaluation to be fairly thorough. While some clarification of the limitations of the results (along the lines of the lengthy responses and discussion with Reviewer 255P) is recommended, this paper overall meets the bar for ICML.